# Repurposing AlphaFold3-like Protein Folding Models for Antibody Sequence and Structure Co-design

**Nianzu Yang**[†1], **Songlin Jiang**[†1], **Jian Ma**[†1], **Huaijin Wu**[1], **Shuangjia Zheng**[3], **Wengong Jin**[*2], **Junchi Yan**[*1]

[1]School of CS & School of AI, Shanghai Jiao Tong University
[2]Khoury College of Computer Sciences, Northeastern University
[3]School of AI & GIFT , Shanghai Jiao Tong University
{yangnianzu,clorf6,majian7,shuangjia.zheng,yanjunchi}@sjtu.edu.cn
w.jin@northeastern.edu

## Abstract

Diffusion models hold great potential for accelerating antibody design, but their performance is so far limited by the number of antibody-antigen complexes used for model training. Meanwhile, AlphaFold3-like protein folding models, pre-trained on a large corpus of crystal structures, have acquired a broad understanding of biomolecular interaction. Based on this insight, we develop a new antigen-conditioned antibody design model by adapting the diffusion module of AlphaFold3-like models for sequence-structure co-diffusion. Specifically, we extend their structure diffusion module with a sequence diffusion head and fine-tune the entire protein folding model for antibody sequence-structure co-design. Our benchmark results show that sequence-structure co-diffusion models not only surpass state-of-the-art antibody design methods in performance but also maintain structure prediction accuracy comparable to the original folding model. Notably, in the antibody co-design task, our method achieves a CDR-H3 recovery rate of 65% for typical antibodies, outperforming the baselines by 87%, and attains a remarkable 63% recovery rate for nanobodies. Our code is available at https://github.com/yangnianzu0515/MFDesign.

## 1 Introduction

Monoclonal antibodies are Y-shaped proteins that specifically recognize and neutralize the pathogens (commonly known as antigens) during immune responses [Janeway et al., 2001]. The binding specificity is determined by their complementarity-determining regions (CDRs), whose sequences and structures exhibit significant variability. Recent work on generative models have shown great potential in designing CDRs that bind to an antigen [Jin et al., 2022a, Luo et al., 2022, Kong et al., 2023a, Martinkus et al., 2024, Zhu et al., 2024]. However, their performance is limited by the scarcity of antibody-antigen complex data, as highlighted by Zhu et al..

To address this issue, we seek to leverage foundation models, such as protein folding models, that have learned general knowledge of biomolecular interaction from a large corpus of general protein interaction data. Our hypothesis is that the general knowledge of protein-protein interaction learned by protein folding models, such as AlphaFold3 (AF3) [Abramson et al., 2024], Chai-1 [Discovery et al., 2024], Boltz-1 [Wohlwend et al., 2024], and Protenix [Chen et al., 2025], are transferrable to the task of modeling of antibody-antigen complexes, a specialized type of protein interaction.

Correspondence author. † denotes equal contribution. This work was partly supported by NSFC (92370201, 62222607) and Shanghai Municipal Science and Technology Major Project under Grant 2021SHZDZX0102.

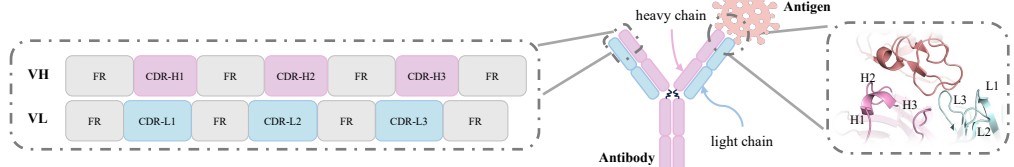

Figure 1: **Illustration of an antibody-antigen complex.** Typical antibodies consist of heavy and light chains, each with a variable domain (VH and VL, respectively) composed of three CDRs and framework regions. CDRs are highly variable and mainly determine the antibody specificity for antigens. In contrast, nanobodies consist of only a heavy chain.

To validate this hypothesis, we need to effectively endow current protein folding models with the additional capability of generating both sequence and structure of an antibody or a protein. Our key observation is that AF3-like folding models already utilize diffusion modules for structure prediction. Indeed, diffusion models [Ho et al., 2020, Song et al., 2021, Karras et al., 2022] have shown various success in the co-design of antibody sequences and structures [Luo et al., 2022, Martinkus et al., 2024, Zhu et al., 2024]. Therefore, we propose to integrate sequence diffusion into the diffusion modules of AF3-like folding models for sequence-structure co-design. We demonstrate the feasibility of this idea using Boltz-1[1] [Wohlwend et al., 2024], an open-source implementation of AF3 provided with training scripts. Specifically, we augment the diffusion module of Boltz-1 with a sequence diffusion head. We implement two approaches for sequence diffusion: a discrete diffusion approach trained in the D3PM-absorbing [Austin et al., 2021a] fashion, and a continuous diffusion approach trained using EDM [Karras et al., 2022] method. After properly aligning sequence and structure diffusion steps, we fine-tune the entire Boltz-1 network using a training set of antibody-antigen complexes. Extensive benchmark results show that the modified AF3-like model outperforms existing baseline methods in antibody sequence and structure co-design tasks, while retaining Boltz-1's structure prediction capability. **In summary, our key contributions are as follows:**

- To the best of our knowledge, we are the first to leverage the general protein interaction knowledge from AF3-like folding foundation models to effectively address the issue of limited antibody-antigen complex data in the context of antibody sequence and structure co-design task.

- Expanding on the structural diffusion framework in AF3-like models, we introduce sequence diffusion alongside a tailored training strategy to enable effective co-design.

- The results show that our modified AF3-like model outperforms existing baselines in antibody sequence and structure co-design tasks, while retaining its original structural prediction capability.

## 2 Related Works

**Diffusion Model.** Diffusion-based generative methods [Song et al., 2021, Ho et al., 2020, Li et al., 2025] have demonstrated remarkable effectiveness across various domains [Li et al., 2023, 2024], such as image and video generation [Epstein et al., 2023, Wu et al., 2024a]. These models can be classified by the type of data they target: continuous or discrete, with the theoretical foundations of continuous diffusion being more developed and mature. Recently, diffusion models have also shed light on new possibilities in the scientific domain. In particular, continuous diffusion methods have excelled in tasks like molecular generation [Guan et al., 2023, Qiang et al., 2023, Yang et al., 2024] and protein structure design [Trippe et al., 2023, Watson et al., 2023], due to their suitability for continuous spatial coordinate data. Meanwhile, discrete diffusion methods have shown promise in handling sequential data, proving effective in designing protein sequences [Gruver et al., 2023a, Ma et al., 2025] and DNA [Avdeyev et al., 2023, Wang et al., 2024], which consist of discrete elements like amino acids and nucleotides.

**Protein Folding and Design Model.** AI models have revolutionized molecule science, including molecular representations [Yang et al., 2022, Yan et al., 2023], molecule generation [Hoogeboom et al., 2022, Wu et al., 2024b], protein docking problem [Wu et al., 2024c] and the protein folding problem. Groundbreaking models like AlphaFold2 [Jumper et al., 2021], trained on the Protein Data Bank (PDB) [Berman et al., 2000], achieved unprecedented accuracy in structure prediction. This

---

[1] ⌂ https://github.com/jwohlwend/boltz

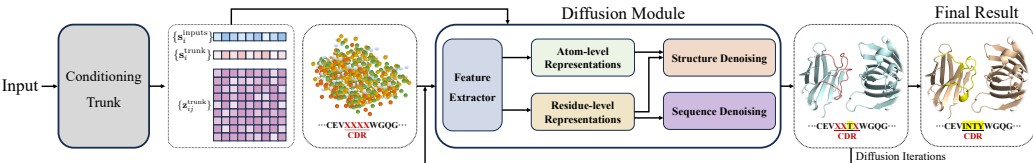

Figure 2: **Overview of our modified AF3-like model.** We build upon AF3-like folding models by integrating sequence diffusion with the existing structure diffusion, enabling the co-diffusion of structure and sequence. The input for sequence denoising network directly reuses the residue token-level representations originally used for structure denoising in AF3-like models. Initially, CDR tokens are marked as `<X>`, which will be specified as one of the 20 amino acid tokens in the final output. Through *replacement* sampling, our method allows the CDR design to be conditioned on fixed co-crystal structures.

progress has continued with models like AlphaFold3 [Abramson et al., 2024], which uses a diffusion-based approach to handle a wider range of molecular interactions, and the powerful RoseTTAFold series [Baek et al., 2021, 2023, 2024]. Building on this success, the field has advanced to the inverse problem: de novo protein design. Generative diffusion models are central to this effort. These models operate in different ways, such as performing diffusion in 3D coordinate space to design protein backbones for tasks like motif scaffolding [Ingraham et al., 2023], or in sequence space to co-design sequence-structure pairs for creating functional, multistate proteins [Lisanza et al., 2024].

**Antibody Design.** Most antibody design efforts focus on the design of CDRs. Traditional methods [Li et al., 2014, Lapidoth et al., 2015, Adolf-Bryfogle et al., 2018] for antibody design primarily rely on energy-based optimization, which can be quite time-consuming. In contrast, recent deep learning-based methods have been proposed to enhance antibody design, broadly divided into discriminative and generative models. Discriminative models [Jin et al., 2022a,b, Kong et al., 2023b,a] typically utilize graph neural networks to extract features from the context, including antigens and antibody framework regions, to predict the most probable CDRs structure and sequence. Generative models [Luo et al., 2022, Martinkus et al., 2024, Zhu et al., 2024], which are mainly diffusion-based, excel at modeling the complex distributions of both structure and sequence. By accurately capturing these distributions, they can sample a more diverse set of antibody candidates from them.

## 3 Methodology

In Sec. 3.1, we first present a detailed description of the task we investigate in this paper. In Sec. 3.2, we introduce key notations by reviewing the workflow of AF3. Finally, in Sec. 3.3, we detail how straightforward modifications to AF3-like models enable support for antibody co-design.

### 3.1 Problem Description

We present a diagram of an antibody-antigen complex in Fig. 1. In this work, we follow the widely adopted setting for antibody design: re-designing CDRs in experimentally resolved co-crystal structures, where all non-CDR regions (antibody framework sequences/structures, antigen sequences/structures, and antibody-antigen binding pose) are given. The objective is to co-design the sequence and structure of CDRs within this predefined context.

Particularly, rather than designing a single CDR at a time, we aim to simultaneously design all CDRs at once, which is a more challenging task [Martinkus et al., 2024]. Within this study, we use the notation system from AF3 to characterize protein complexes, employing $\{\mathbf{k}_i\}$ and $\{\mathbf{x}_l\}$ to specify the amino acid (*a.k.a.* residue) composition and structure of the complex, respectively. $\mathbf{k}_i$ is a one-hot encoding that represents the type of each amino acid token (20 standard amino acid types + `<X>`). The `<X>` token is originally used in AF3-like folding models to indicate positions where the residue type is unknown, for which only the backbone atom coordinates will be predicted. Each element in $\{\mathbf{x}_l\}$ represents the three-dimensional coordinates of an atom.

We use the `<X>` token to represent residues within the CDRs that need to be designed. In the final output, each `<X>` token is replaced with a specific residue from the 20 standard types, accompanied by their corresponding atomic coordinates, thus achieving the goal of the antibody co-design.

## 3.2 Preliminary: AlphaFold3-like Folding Models

AF3-like methods are structured as conditional diffusion models, consisting mainly of two components: the conditioning trunk and the diffusion module. These models also feature a confidence module, but it is trained independently and is not the primary focus here, so we do not discuss it. Below, notations with subscript $i/ij$ are token-level representations, while $l/lm$ are atom-level. The modules introduced in the original AF3 paper are **highlighted** in the following discussion, and corresponding details can be referenced in the supplementary materials of AF3[2].

**Conditioning Trunk.** This trunk processes raw inputs denoted as $\{\mathbf{f}^*\}$, such as the protein sequences to be predicted, MSA results, and structural template information. Initially, these inputs pass through an **InputEmbedder**, which performs basic encoding to obtain input features $\{\mathbf{s}_i^{\text{inputs}}\}$. These initial input features are used to derive the preliminary single and pair representations. The single and pair representations are then updated iteratively through a stack of three modules: **TemplateModule**, **MSAModule**, and **PairFormerStack**. Finally, the input features $\{\mathbf{s}_i^{\text{inputs}}\}$ obtained at the beginning, along with the iteratively updated single and pair representations, denoted as $\{\mathbf{s}_i^{\text{trunk}}\}$ and $\{\mathbf{z}_{ij}^{\text{trunk}}\}$ respectively, are fed to the following diffusion module to condition the structural diffusion process.

**Structure Diffusion.** AF3 adopts EDM [Karras et al., 2022] for structure diffusion training. Notably, EDM differs from typical diffusion models [Ho et al., 2020, Song et al., 2021] that sample a timestep to determine the noise level based on a pre-defined noise schedule during training. Instead, EDM directly samples the noise level by drawing the logarithm of the noise intensity from a defined Gaussian distribution.

Specifically, during AF3's training, the noise level $\sigma$ is sampled from the distribution $\sigma_{\text{data}} \cdot \exp(-1.2 + 1.5 \cdot \mathcal{N}(0, 1))$ and is fed, along with the three outputs from the previous trunk, into the **DiffusionConditioning** module. In this module, relative position encoding between residue tokens is incorporated into $\{\mathbf{z}_{ij}^{\text{trunk}}\}$ to facilitate both token-level and atom-level self-attention [Vaswani, 2017] used later, resulting in the output $\{\mathbf{z}_{ij}\}$. Besides, within this module, the sampled noise level is embedded using random Fourier features [Tancik et al., 2020] and then integrated with $\{\mathbf{s}_i^{\text{inputs}}\}$ and $\{\mathbf{s}_i^{\text{trunk}}\}$ to construct $\{\mathbf{s}_i\}$.

Next, following EDM, AF3 scales the noisy atom coordinates $\{\mathbf{x}_l^{\text{noisy}}\}$ to unit variance, which gives $\{\mathbf{r}_l^{\text{noisy}}\}$. $\{\mathbf{r}_l^{\text{noisy}}\}$, $\{\mathbf{s}_i^{\text{trunk}}\}$ and $\{\mathbf{z}_{ij}\}$ are further processed in the **AtomAttentionEncoder**. Initially, this module yields three atom-level representations: $\{\mathbf{q}_l\}$, $\{\mathbf{c}_l\}$, $\{\mathbf{p}_{lm}\}$. These representations are all based on atomic features from the raw inputs and incorporate additional information from $\{\mathbf{r}_l^{\text{noisy}}\}$, $\{\mathbf{s}_i^{\text{trunk}}\}$ and $\{\mathbf{z}_{ij}\}$, respectively. Then, sequence-local atom attention is applied to update $\{\mathbf{q}_l\}$ using $\{\mathbf{c}_l\}$ and $\{\mathbf{p}_{lm}\}$. The updated $\{\mathbf{q}_l\}$ are then aggregated to form the token-level representations $\{\mathbf{a}_i\}$.

Subsequently, $\{\mathbf{s}_i\}$ is used to update $\{\mathbf{a}_i\}$. The updated $\{\mathbf{a}_i\}$, along with outputs $\{\mathbf{a}_i\}$ and $\{\mathbf{z}_{ij}\}$ from the previous **DiffusionConditioning** module, is fed into the **DiffusionTransformer** module to perform full self-attention on the token level. This module further updates $\{\mathbf{a}_i\}$ and layer normalization is applied to $\{\mathbf{a}_i\}$.

The residue-level representations $\{\mathbf{a}_i\}$, together with the atom-level representations $\{\mathbf{q}_l\}$, $\{\mathbf{c}_l\}$ and $\{\mathbf{p}_{lm}\}$ are then inputted into the **AtomAttentionDecoder** module, which yields $\{\mathbf{r}_l^{\text{update}}\}$. $\{\mathbf{r}_l^{\text{update}}\}$ is rescaled back and combined with the input noisy structure $\{\mathbf{r}_l^{\text{noisy}}\}$, in line with EDM, to obtain the final predicted denoised structure $\{\mathbf{x}_l^{\text{denoised}}\}$.

The structure diffusion training process outlined above is encapsulated within the **DiffusionModule** of AF3, as formalized in Algorithm 20 of AF3's supplementary materials. In summary, this process uses the outputs from the trunk, in conjunction with the raw inputs, to further construct atom-level and token-level representations. These representations are then fed to the structure denoising network for recovery.

For inference, AF3 adopts the EDM stochastic sampler and from a sequence of 200 decreasing levels defined by:

$$\left\{ \sigma_{\text{data}} \cdot \left( \sigma_{\max}^{1/\rho} + \frac{t}{200 - 1} \cdot \left( \sigma_{\min}^{1/\rho} - \sigma_{\max}^{1/\rho} \right) \right)^{\rho} \,\middle|\, t = 0, 1, \ldots, 199 \right\},$$

---

[2] ⤤ Click here to access the supplementary materials of AF3.

---

**Algorithm 1** Sequence and Structure Co-diffusion Module: **CoDiffusionModule**

---

**Input:** noisy structure $\{\mathbf{x}_l^{\text{noisy}}\}$, noisy sequence $\{\mathbf{k}_i^{\text{noisy}}\}$, noise level for structure $\sigma$, raw inputs $\{\mathbf{f}^*\}$,
trunk's outputs $\left\{\{\mathbf{s}_i^{\text{inputs}}\}, \{\mathbf{s}_i^{\text{trunk}}\}, \{\mathbf{z}_{ij}^{\text{trunk}}\}\right\}$, hyper-parameter $\{\sigma_{\text{data}}\}$;

1: $\{\mathbf{s}_i^{\text{replaced}}\} \leftarrow$ substitute the specified dimensions within $\{\mathbf{s}_i^{\text{inputs}}\}$ with $\{\mathbf{k}_i^{\text{noisy}}\}$
   *# Pre-conditioning for diffusion*

2: $\{\mathbf{s}_i\}, \{\mathbf{z}_{ij}\} = \textbf{DiffusionConditioning}\left(\sigma, \{\mathbf{f}^*\}, \{\mathbf{s}_i^{\text{replaced}}\}, \{\mathbf{s}_i^{\text{trunk}}\}, \{\mathbf{z}_{ij}^{\text{trunk}}\}, \sigma_{\text{data}}\right)$ ▶ Refer to Algorithm 21 in AF3
   *# Scale coordinates with approximately unit variance*

3: $\mathbf{r}_l^{\text{noisy}} = \mathbf{x}_l^{\text{noisy}}/\sqrt{\sigma^2 + \sigma_{\text{data}}^2}$
   *# Sequence-local atom attention and aggregation to coarse-grained tokens*

4: $\{\mathbf{a}_i\}, \{\mathbf{q}_l\}, \{\mathbf{c}_l\}, \{\mathbf{p}_{lm}\} = \textbf{AtomAttentionEncoder}\left(\{\mathbf{f}^*\}, \{\mathbf{r}_l^{\text{noisy}}\}, \{\mathbf{s}_i^{\text{trunk}}\}, \{\mathbf{z}_{ij}\}\right)$ ▶ Refer to Algorithm 5 in AF3
   *# Full self-attention on token level*

5: $\mathbf{a}_i \mathrel{+}= \text{LinearNoBias}(\text{LayerNorm}(\mathbf{s}_i))$

6: $\{\mathbf{a}_i\} = \textbf{DiffusionTransformer}\left(\{\mathbf{a}_i\}, \{\mathbf{s}_i\}, \{\mathbf{z}_{ij}\}\right)$ ▶ Refer to Algorithm 23 in AF3

7: $\mathbf{a}_i = \text{LayerNorm}\left(\mathbf{a}_i\right)$
   *# Structure denoising*

8: $\left\{\mathbf{r}_l^{\text{update}}\right\} = \textbf{AtomAttentionDecoder}\left(\{\mathbf{a}_i\}, \{\mathbf{q}_l\}, \{\mathbf{c}_l\}, \{\mathbf{p}_{lm}\}\right)$ ▶ Refer to Algorithm 6 in AF3

9: $\mathbf{x}_l^{\text{denoised}} = \sigma_{\text{data}}^2/\left(\sigma_{\text{data}}^2 + \sigma^2\right) \cdot \mathbf{x}_l^{\text{noisy}} + \sigma_{\text{data}} \cdot \sigma/\sqrt{\sigma_{\text{data}}^2 + \sigma^2} \cdot \mathbf{r}_l^{\text{update}}$
   *# Sequence denoising*

10: $\{\mathbf{k}_i^{\text{denoised}}\} = \textbf{TokenDenoiser}\left(\{\mathbf{a}_i\}\right)$

**Output:** denoised structure $\{\mathbf{x}_l^{\text{denoised}}\}$, denoised sequence $\{\mathbf{k}_i^{\text{denoised}}\}$;

---

where $\sigma_{\text{data}}$, $\sigma_{\text{max}}$, $\sigma_{\text{min}}$ and $\rho$ are hyper-parameters.

## 3.3 Our Method

The input includes residue tokens from antibodies and antigens. For the antibody CDRs that need to be designed, these positions are filled with <X> tokens. Considering the discrete nature of residue tokens as categorical variables, along with the resemblance of the input format to masked language models (MLMs) like BERT [Devlin et al., 2019] — where <X> is analogous to the <MASK> tokens used in MLMs — the D3PM-absorbing [Austin et al., 2021a] method, inspired by MLMs, intuitively aligns well with this task. Therefore, we choose to use D3PM-absorbing for sequence diffusion.

During the training of diffusion-based co-design methods, it is required to corrupt the sequence and structure to a matching degree, which can be achieved by sampling them at the same timestep. However, as noted in Sec. 3.2, AF3-like models perform structure diffusion training by directly sampling noise levels, unlike D3PM-absorbing, which uses timestep sampling. Furthermore, to the best of our knowledge, there is no established theory for training discrete diffusion without sampling timesteps during so far. Aligning sequence and structure diffusion is therefore challenging.

**Continuous diffusion as an alternative for sequence**. It is worth noting that Hoogeboom et al. propose a method for applying continuous diffusion to discrete data by adding Gaussian noise to their one-hot encodings in training. This allows sequence to be trained also using the EDM method. In this way, sequence diffusion can readily align with structure diffusion training in AF3. For the generation of discrete data, this method tends to be less effective than discrete diffusion [Austin et al., 2021a, Gruver et al., 2023a], as supported by results in Sec. 4.1, which we have also implemented. However, our results also reveal its unique advantage for structure prediction. Since this paper primarily focuses on the additional implementation of sequence design compared to AF3, which involves discrete data, we have concentrated on discussing the method using D3PM-absorbing for sequence diffusion in the following main paper. The implementation of the continuous version, being relatively more straightforward and simpler, is provided in Appendix B.4.

**Structure & sequence diffusion alignment.** Although the training method for structure diffusion directly samples noise levels, we notice that AF3 uses a predefined sequence of 200 decreasing noise levels during sampling. This approach closely resembles the concept of discrete timesteps. Thus, we consider modifying structure diffusion training to also sample noise levels only from these 200 noise levels. This allows us to create a predefined noise scheduler for training, where we sample a timestep and retrieve the corresponding noise level according to this scheduler. Accordingly, we set the number of diffusion steps for sequence to 200, thus enabling easy alignment. Since these same 200 levels are used during both training and inference of the structure diffusion, the approach shifts towards a style resembling the classic DDPM's discrete time training [Ho et al., 2020]. Empirical results suggest that fine-tuning with this new structure diffusion approach preserves the model's ability to accurately predict complex structures.

**Instantiation of discrete sequence diffusion.** Diffusion is applied specifically to CDRs. <X> is naturally suited as the absorbing state for residues in CDRs. The forward process is characterized by a discrete transition matrix that determines the probability of a token mutating into <X>. The corresponding prior is a point mass on the sequence of which the CDRs are made up entirely of <X>. On top of AF3, we introduce an additional module named **TokenDenoiser**. Since we need to make predictions at token-level, token-level representations are required for **TokenDenoiser**. Recalling from Sec 3.2, the AF3's structure diffusion process already drives token-level representations $\{\mathbf{a}_i\}$, which are obtained via full self-attention on token level and serve the purpose of structure denoising. Here we choose to directly reuse $\{\mathbf{a}_i\}$ as the input to **TokenDenoiser**. The **TokenDenoiser** is implemented simply as a basic MLP in this paper, consisting of LINEAR layers connected by GELU activation function.

Next, we detail how to input the corresponding noisy sequence, denoted as $\{\mathbf{k}_i^{\text{noisy}}\}$, into the diffusion module at each sampled timestep during training. As mentioned in Sec. 3.2, $\{\mathbf{s}_i^{\text{inputs}}\}$ from the trunk serves as a basic encoding of the raw input, where part of its continuous dimensions is actually the one-hot encoding to the tokens. Thus, we can easily achieve the input of the noisy sequence at sampled timestep by directly replacing these dimensions with $\{\mathbf{k}_i^{\text{noisy}}\}$, resulting in $\{\mathbf{s}_i^{\text{replaced}}\}$. Then, $\{\mathbf{s}_i^{\text{replaced}}\}$ is fed into our structure and sequence co-diffusion module and its subsequent usage remains consistent with that of $\{\mathbf{s}_i^{\text{inputs}}\}$. We denote the restored sequence as $\{\mathbf{k}_i^{\text{denoised}}\}$. Note that except for $\{\mathbf{s}_i^{\text{inputs}}\}$, all these notations are associated with a specific timestep $t$. However, for simplicity, we omit $t$ in the notations presented here.

We fully fine-tune the AF3-like models by combining the original objective of AF3 for structure prediction with an additional objective specifically for sequence diffusion. This additional objective is to maximize the likelihood of the denoising process on the ground truth CDRs sequences, thereby learning the optimal parameters for the **TokenDenoiser**. Our sequence sampling follows the D3PM-absorbing standard method, as used by Gruver et al.. To allow our sampling to be conditioned on the given co-crystal structures and redesign CDRs, we employ the *replacement* sampling technique. Orginally from image inpainting domain [Lugmayr et al., 2022], this technique has been applied in molecule and protein generation tasks to condition the sampling on certain motifs [Schneuing et al., 2024, Trippe et al., 2023]. Refer to Appendix B for more details on our training and sampling.

Our method is outlined in Fig. 2. We extend the AF3's **DiffusionModule** with sequence diffusion to enable structure and sequence co-design, which we call **CoDiffusionModule**, formalized in Alg. 1. It follows a feature extraction process akin to the **DiffusionModule**, but reuses the final obtained token-level representations for sequence design.

**Training strategy.** Similar to AF3 models that use token cropping for training, we also implement this technique. Our training process is divided into four stages as follows:

- **1st-stage:** We only input antibody sequences, using a maximum token size of 256 to accommodate the VH and VL sequences. This allows for a larger batch size, enabling the model to quickly learn to design CDRs and predict antibody structures based only on antibody sequences.

- **2nd-stage & 3rd-stage:** We continue to input full VH and VL sequences while randomly selecting tokens from the antigen epitope and the nearby regions to introduce the antigen context. Both stages follow the same cropping method detailed in Alg. S2, differing only in maximum token size: 384 in the second stage and 512 in the third.

- **4th-stage:** We perform full-parameter fine-tuning, which may affect the model's ability to predict structure learned during pre-training. Besides, considering the first three stages focus on predicting structures around the antibody and nearby antigen regions, the model might underperform when predicting structures of antigen regions far from the antibody. To address this, we implement a sampling strategy where there is a 50% probability of selecting tokens from regions around the antibody and a 50% probability of selecting tokens from any area within the complex. This approach helps the model retains its newly acquired antibody co-design capability while also preserving its pre-trained ability to predict complex structures. The maximum token size for this stage is kept at 512.

**Avoiding data leakage.** Even when the CDRs in the antibody query sequences for MSA are filled with the unknown token <X>, the MSA results contain a significant number of sequences that, while not exactly identical to the antibody sequences, have sequences in positions corresponding to the

Table 1: Comparison of simultaneously designed CDRs for typical antibodies, with the **best** and the second-best highlighted.

| CDR | Method | AAR ↑ | RMSD ↓ | CDR | Method | AAR ↑ | RMSD ↓ |
|---|---|---|---|---|---|---|---|
| H1 | DiffAb | 60.92% | 1.52Å | L1 | DiffAb | 56.72% | 1.43Å |
| | dyMEAN | 70.31% | 1.65Å | | dyMEAN | 73.06% | 1.58Å |
| | MFDesign-C | 74.89% | 1.48Å | | MFDesign-C | 82.98% | 1.41Å |
| | MFDesign-D | 74.95% | 1.61Å | | MFDesign-D | 82.98% | 1.65Å |
| H2 | DiffAb | 33.53% | 1.44Å | L2 | DiffAb | 55.08% | 1.21Å |
| | dyMEAN | 66.38% | 1.47Å | | dyMEAN | 76.04% | 1.23Å |
| | MFDesign-C | 65.59% | 1.26Å | | MFDesign-C | 88.84% | 1.00Å |
| | MFDesign-D | 67.54% | 1.44Å | | MFDesign-D | 87.81% | 1.15Å |
| H3 | DiffAb | 22.26% | 4.29Å | L3 | DiffAb | 43.03% | 1.80Å |
| | dyMEAN | 34.69% | 6.15Å | | dyMEAN | 52.69% | 1.59Å |
| | MFDesign-C | 63.13% | 3.54Å | | MFDesign-C | 78.93% | 1.55Å |
| | MFDesign-D | 65.04% | 3.71Å | | MFDesign-D | 80.15% | 1.69Å |

Table 2: Results on typical antibodies with simultaneously designed CDRs, highlighting the **best** and the second-best results.

| Method | Loop-AAR ↑ | Loop-RMSD ↓ | IMP ↑ |
|---|---|---|---|
| DiffAb | 15.86% | 5.03Å | 53.35% |
| dyMEAN | 20.80% | 7.84Å | 5.60% |
| MFDesign-C | 61.71% | 4.13Å | 65.37% |
| MFDesign-D | 63.38% | 4.28Å | 59.16% |

antibody CDRs highly consistent with the CDRs ground-truth sequences. This can cause the model to learn to predict CDR sequences directly from the MSA results, leading to data leakage. To address this, in both our training and inference, we exclude sequences where the regions corresponding to the CDRs in the query antibody sequence have a similarity to the CDRs that exceeds a threshold, as detailed in Appendix A.4. This processing is to ensure fair benchmarking. In practical antibody design, filtering is not required.

AF3-like folding models inherently support the input of `<X>` tokens. For these unknown residues, they only sample and denoise the backbone atom coordinates. In line with this design, we exclusively generate the backbone atom coordinates for the CDRs. Subsequently, following Luo et al. and Zhu et al., we perform side-chain packing and structure relaxation using PyRosetta [Chaudhury et al., 2010] to obtain final all-atom structure.

# 4 Experiments

Due to the unavailability of AF3's training scripts, we opt to make modifications based on Boltz-1 in this work, which is a faithful reimplementation by Wohlwend et al. that follows the same variable and function definitions as AF3. Our method is termed **MFDesign** in subsequent sections, which stands for the **M**odified **F**olding model for the co-**D**esign of antibody sequence and structure.

## 4.1 Antibody Sequence and Structure Co-design

**Dataset.** The data for evaluation comes from the Structural Antibody Database (SAbDab) [Dunbar et al., 2014]. Boltz-1 itself is pre-trained on PDB structures released before the same cut-off date as AlphaFold3, *i.e.*, September 30, 2021. In previous studies [Luo et al., 2022, Kong et al., 2023a, Martinkus et al., 2024, Zhu et al., 2024], their experimental setups do not consider the release date when splitting data. To prevent data leakage, meaning the test data should not have been seen during Boltz-1's pre-training phase, we cannot directly adopt the splits used by previous works. Instead, we need to consider the release date when partitioning the data. We first follow the approach of DiffAb [Luo et al., 2022] and AbDiffuser [Martinkus et al., 2024] by using MMSeq2 [Steinegger and Söding, 2017] to cluster antibodies according to CDR-H3 sequences at 50% sequence identity. These clusters are then divided into training, validation, and test sets in an 9:0.5:0.5 ratio, ensuring that all samples released before the cut-off date are included in the training set. This results in 5,843 training samples, 187 validation samples, and 204 test samples, with the test set comprising 161 regular antibodies and 43 nanobodies. Our test set is much larger than those in previous studies, offering a more robust demonstration of the models' ability to generalize. Due to GPU memory limitations, we ultimately removed samples with more than 2000 amino acids. Moreover, it's worth noting that some prior works [Jin et al., 2022b, Kong et al., 2023a, Zhu et al., 2024] employ the RAbD Benchmark [Adolf-Bryfogle et al., 2018], which includes 60 diverse complexes, to test antibody generation performance. However, since these data are released in the PDB before the cut-off date and Boltz-1 has already seen them, we do not adopt the RAbD Benchmark for evaluation to ensure fair comparisons. We provide our detailed data processing procedure in Appendix A to establish a standardized pipeline, which aims to assist future studies in replicating or building upon our work.

**Baselines.** We focus on the more challenging setting of designing all CDRs at once [Martinkus et al., 2024], rather than designing a single CDR at a time. Consequently, we do not compare our approach with those methods that are limited to designing a single CDR at a time. Although both

Table 3: Comparison of generated nanobodies with simultaneously designed CDRs, assessed based on AAR and RMSD for CDR-H1, H2, and H3, as well as Loop-AAR, Loop-RMSD, IMP metrics, with the **best** and the second-best results highlighted. Note that we do not compare against dyMEAN here, as its source code does not support processing nanobodies.

| Method | CDR-H1 | | CDR-H2 | | CDR-H3 | | Loop-AAR / % | Loop-RMSD / Å | IMP / % |
|---|---|---|---|---|---|---|---|---|---|
| | AAR / % | RMSD / Å | AAR / % | RMSD / Å | AAR / % | RMSD / Å | | | |
| DiffAb | 45.17 | 2.52 | 31.11 | **1.54** | 15.39 | 4.71 | 10.51 | 5.50 | 30.93 |
| MFDesign-C | 65.71 | **2.48** | 55.60 | 1.62 | 61.40 | **3.71** | **61.01** | **4.26** | 43.14 |
| MFDesign-D | **67.34** | 2.62 | **58.93** | 1.78 | **63.51** | 3.84 | 60.80 | 4.41 | 38.26 |

AbDiffuser [Martinkus et al., 2024] and AbX [Zhu et al., 2024] are capable of designing all CDRs simultaneously, we exclude these methods from our comparison due to the unavailability of available source code for AbDiffuser and the absence of training scripts for AbX [Zhu et al., 2024]. Ultimately, we choose **DiffAb** [Luo et al., 2022] and **dyMEAN** [Kong et al., 2023a] as our baselines because they serve as representative methods for the generative and discriminative categories of antibody co-design methods, respectively. For our method, we provide results for sequence diffusion implemented with both discrete and continuous diffusion. We denote the discrete version of the model as **MFDesign-D** and the continuous version as **MFDesign-C**.

**Metrics.** We adopt the following metrics to evaluate all methods: (1) **AAR** (%): measures the accuracy of the generated sequences of CDRs by comparing amino acid identity with native sequences; (2) **RMSD** (Å): calculates the root-mean-square deviation of $C_\alpha$ atom coordinates between generated and native CDRs. Following Zhu et al., we introduce additional metrics for the middle loop residues within CDR-H3, which primarily contribute to antigen binding: (3) **Loop-AAR** (%): applies AAR specifically to the middle loop residues of CDR-H3, assessing sequence recovery in this key region; (4) **Loop-RMSD** (Å): uses RMSD to evaluate the structural accuracy of the middle loop residues within CDR-H3. Furthermore, we use: (5) **IMP** (%): is the percentage of designed antibodies with enhanced binding energy ($\Delta G$) compared to the original, assessed with Rosetta's *InterfaceAnalyzer* tool [Alford et al., 2017]. We also evaluate the binding energy $\Delta G$ of designed antibodies, with full results detailed in Appendix E.3 due to space constraints. For generative-based methods, DiffAb and our method, we generate 20 candidates per sample and report the averaged results.

For a fair comparison, we follow the same experimental setting as previous works [Luo et al., 2022, Kong et al., 2023a, Martinkus et al., 2024, Zhu et al., 2024], which involves re-designing CDRs within given co-crystal structures. In this setup, the sequences and structures of all non-CDR regions, including the framework and antigen, along with the antibody-antigen binding conformation, are provided. By using the *replacement* sampling technique mentioned in Sec. 3.3, our method enables CDRs re-design based on the given co-crystal structures. The results are presented below.

**Results on Typical Antibodies.** Table 1 and Table 2 present the results of simultaneously designed CDRs for typical antibodies, fully demonstrating the superiority of our method. From Table 1, we see that MFDesign-C achieves the best results in RMSD across all CDRs. Its performance in AAR is also superior to the baselines, except for CDR-H2. MFDesign-D, while not surpassing our own continuous version on CDR-L2, outperforms the baselines in AAR across the remaining CDRs by a notable margin. This reflects the suitability of each model for different data types: coordinates are continuous, so the continuous version generally performs better in RMSD; sequences are discrete, so the discrete version generally performs better in AAR. Notably, for the more challenging CDR-H3, our method improves AAR by 87.49% and reduces RMSD by 17.48% compared to the best baseline. Table 2 shows the results when further ignoring the relatively conserved regions at both ends of CDR-H3 and focusing only on the highly variable middle loop, as these residues are the primary contributors to antigen binding [Kong et al., 2023a, Zhu et al., 2024]. In this case, the best performance in Loop-AAR among the baselines, achieved by dyMEAN, is only 20.80%, while MFDesign-D achieves an accuracy of 63.38%, representing a 204.71% improvement. In terms of Loop-RMSD, MFDesign-D attains a 17.89% reduction compared to the best baseline DiffAb. Besides, our method reaches 65.37% in the IMP metric. We attribute the superior performance of our method to the broad knowledge of protein interactions originally acquired by AF3-like model during pre-training. Moreover, the AF3-like model themselves already have good structure prediction capabilities, which may also explain the effectiveness of MFDesign in RMSD-related metrics. To further validate the robustness and generalization of our approach, we additionally evaluated our model on the widely used RAbD benchmark, which serves as a standard dataset in previous antibody design studies. As shown in Table S8 and Table S9, our method consistently outperforms all baseline methods across multiple metrics, confirming its superior performance and general applicability.

Table 4: Performance comparison on the structure prediction task, evaluated by RMSD, Loop-RMSD, and TM-score.

| Given CDR? | Fine-tuned? | RMSD ↓ | Loop-RMSD ↓ | TM-score ↑ |
|:---:|:---:|:---:|:---:|:---:|
| ✗ | ✗ | 2.21Å | 5.57Å | 91.84% |
| ✗ | ✓ | 1.73Å | 4.90Å | 93.88% |
| ✓ | ✗ | 1.39Å | 3.53Å | 95.45% |
| ✓ | ✓ | 1.65Å | 4.55Å | 94.25% |

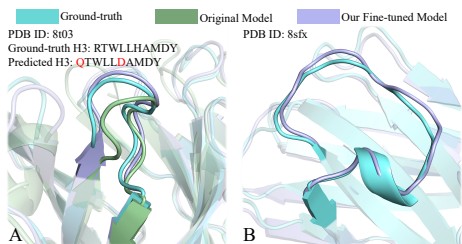

Figure 3: **Examples of predicted structures.**

**Results on Nanobodies.** Nanobodies, *a.k.a.* single-domain antibodies, consist of only a single heavy chain and are gaining increasing attention due to thier favorable properties, such as high solubility and stability [Bannas et al., 2017]. Their structure is simpler, but because they lack a light chain to assist in antigen binding, their CDR-H3 region tends to be longer compared to typical antibodies and nanobody-antigen complex data is scarcer than typical antibodies. This longer CDR-H3 aids in antigen binding but also makes it more challenging to design. In this study, we validate the effectiveness of our method for designing nanobodies, where the labeled data is even more less, and summarize the results in Table 3. We observe that for all methods, performance decreases compared to typical antibodies, which may be due to the limited data availability. The baseline DiffAb shows a significant drop, especially in nanobody CDR-H3, where its AAR is only 15.39%, a 30.86% decrease. In contrast, both the continuous and discrete versions of our method maintain an AAR above 60% for CDR-H3, with Loop-AARs also above 60%. In addition, to ensure the credibility of our methods, we report the standard error of the mean (SEM) for all antibody and nanobody results (Tables S10–S12). We also include the average inference time per sample for different methods, with detailed analysis provided in Appendix E.10.

## 4.2 Structure Prediction

In this section, we evaluate our method in the structure prediction task, *a.k.a.* protein folding problem. Unlike in Sec. 4.1, we do not use impainting for sampling for this task because revealing any ground-truth structure information is not allowed here. We observe that the origin Boltz-1 struggles with accurately predicting antibody-antigen binding sites, a common issue among existing structure prediction models. This limitation becomes evident when Boltz-1 is tasked with predicting complex structures for antibody-antigen pairs from the test set in Sec. 4.1. For each sample, the model generates five predicted structures, and the average TM-score across these predictions is only 0.62. As our model is fine-tuned from Boltz-1, it inherits this limitation.

Thus, we choose to predict antibody structures alone, a relatively simpler task, yet still challenging, particularly in accurately predicting the CDR-H3 region. We experiment with two input types: one where the CDRs are represented as <X>, and another where the CDRs are provided. We compare our fine-tuned Boltz-1 model with the original, non-fine-tuned model. For the first type, we apply filtering to the MSA results. For the second type, we do not filter the MSA results. We adopt the test set used in Sec. 4.1. We use the model implemented in discrete sequence diffusion here for evaluation.

**Metrics.** We utilize the following set of metrics for detailed evaluation: (1) **RMSD** (Å): measures the root-mean-square deviation of all $C_\alpha$ atom coordinates after aligning the entire protein structure, providing an overall measure of structural accuracy; (2) **Loop-RMSD** (Å): calculates the RMSD for the $C_\alpha$ atoms specifically in the middle loop of the CDR-H3 region after whole-structure alignment, which is particularly challenging to predict; (3) **TM-score**: ranges from 0 to 1, with higher scores indicating better similarity between the predicted and reference structures. We generate 20 predictions for each sample and report the averaged results.

**Structure Prediction of Antibody** As shown in Table 4, in scenarios where the input antibody's CDRs are unknown, our fine-tuned model consistently outperforms the original across all three metrics. In Fig. 3A, we demonstrate a case where CDRs are not present in the input. We see that the original Boltz-1 and our fine-tuned model can both accurately predict the framework regions of this sample. However, for the CDRs, the original model's prediction is less precise. Our model, despite starting with unknown residues, can predict CDRs sequences during the sampling process. As the sequence becomes clearer, this allows for more accurate CDRs structure predictions. This suggests that our method could provide valuable insights into what the functional structure of an antibody might look like, when only the framework regions sequences are given.

Table 5: Comparison of simultaneously designed CDRs for typical antibodies using different discrete diffusion variants, with the **Best** and second-best results are highlighted.

| Method | H1 | | L1 | | H2 | | L2 | | H3 | | L3 | | Loop-AAR / % | Loop-RMSD / Å | IMP / % |
|---|---|---|---|---|---|---|---|---|---|---|---|---|---|---|---|
| | AAR / % | RMSD / Å | AAR / % | RMSD / Å | AAR / % | RMSD / Å | AAR / % | RMSD / Å | AAR / % | RMSD / Å | AAR / % | RMSD / Å | | | |
| MFDesign-C | 74.89 | 1.48 | 82.98 | 1.41 | 65.59 | 1.26 | 88.84 | 1.00 | 63.13 | 3.54 | 78.93 | 1.55 | 61.71 | 4.13 | 66.24 |
| MFDesign-D (absorb) | 74.95 | 1.61 | 82.98 | 1.65 | 67.54 | 1.44 | 87.81 | 1.15 | 65.04 | 3.71 | 80.15 | 1.69 | 63.38 | 4.28 | 59.66 |
| MFDesign-D (uniform) | 71.82 | 1.41 | 77.09 | 1.38 | 55.74 | 1.29 | 82.47 | 0.98 | 50.15 | 3.70 | 73.06 | 1.55 | 48.99 | 4.30 | 64.72 |

Table 6: Nanobody design results on CDRs, loop regions, and overall binding improvement (IMP).

| Method | CDR-H1 | | CDR-H2 | | CDR-H3 | | Loop-AAR / % | Loop-RMSD / Å | IMP / % |
|---|---|---|---|---|---|---|---|---|---|
| | AAR / % ↑ | RMSD / Å ↓ | AAR / % ↑ | RMSD / Å ↓ | AAR / % ↑ | RMSD / Å ↓ | | | |
| MFDesign-C | 65.71 | 2.48 | 55.60 | 1.62 | 61.40 | 3.71 | 61.01 | 4.26 | 42.67 |
| MFDesign-D (absorb) | 67.34 | 2.62 | 58.93 | 1.78 | 63.51 | 3.84 | 60.80 | 4.41 | 38.26 |
| MFDesign-D (uniform) | 60.08 | 2.51 | 46.29 | 1.65 | 45.45 | 3.60 | 43.88 | 4.14 | 43.84 |

With the CDRs provided, we observe that the results are improved for both compared to when CDRs are absent. Our fine-tuned model performs worse than the original model on three metrics. However, the gap is slight, with the TM-score decreasing by only 1.26%. This indicates that our fine-tuning retains a structure prediction capability comparable to that of the original model. This slight decline in performance might be due to changes in the training method for structural diffusion. Additionally, because we apply full parameter fine-tuning, it might change the feature space learned during pre-training that was more beneficial for structure prediction. The added sequence prediction task may introduce gradients that conflict with structure prediction, leading to decreased performance in the latter. In Fig. 3B, we present the prediction of the fine-tuned model for a nanobody, whose CDR-H3 is longer and thus more challenging to predict accurately than that of a typical antibody. However, our fine-tuned model successfully predicts the structure of the CDR-H3 for this sample, further exemplifying our method's good structure prediction ability.

**Structure Prediction of General Protein Complex** To evaluate the generalizability of MF-Design beyond antibody-specific applications, we assessed its structure prediction performance on 128 pure protein complexes from the Boltz-1 test set and compared it against the original Boltz-1 model. Among our variants, MFDesign-C was selected due to its superior structure modeling capabilities. As shown in Table S13, MFDesign-C achieves comparable accuracy to Boltz-1 across key structural metrics, including TM-score, RMSD, and DockQ. Given that MF-Design was fine-tuned specifically for antibody modeling, a moderate decrease in performance on general protein complexes is anticipated, reflecting the trade-off introduced by domain-specific adaptation.

### 4.3 Ablation of the Discrete Methods

We selected D3PM-absorb as the diffusion kernel for sequence generation due to its natural fit for discrete amino acid sequences. Its absorbing noise process mirrors masked language modeling, a proven technique for sequence-based tasks. This choice is supported by prior work demonstrating its superiority in text generation [Austin et al., 2021b] and its successful application in recent antibody design models [Gruver et al., 2023b].

To validate this decision, we conducted an ablation study comparing D3PM-absorb with D3PM-uniform. As shown in the results Table 5 and 6, D3PM-absorb achieved significantly better sequence reconstruction (Average Accuracy of Recovery), while D3PM-uniform offered no clear structural benefits. This confirms that D3PM-absorb is the more effective and suitable choice for our model.

## 5 Conclusion

This paper proposes an antigen-conditioned antibody design model by extending AlphaFold3-like models with sequence-structure co-diffusion. By utilizing their acquired understanding of biomolecular interactions during pre-training, our method effectively overcomes the challenge posed by the limited availability of labeled antibody-antigen complex data. Our modified AF3-like model achieves superior performance over existing baselines in the co-design of antibody sequence and structure, while maintaining structure prediction accuracy comparable to the original AF3 model.

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

# Appendix

## Contents

# A  Data Preprocess

Here, we present a detailed description of our data processing pipeline for the Structural Antibody Database (**SAbDab**) [Dunbar et al., 2014], with the goal of establishing a standardized approach that facilitates reproducibility and enables future research endeavors in this field. As a comprehensive repository of antibody structures that undergoes weekly updates, SAbDab provides an invaluable resource for structural antibody research. The data used in this work are collected from SAbDab up to November 27, 2024. The full data processing scripts and processed data will be released upon acceptance.

## A.1  Preliminary Data Filtering

Based on the summary file in TSV format provided by SAbDab, we only include samples whose annotated `antigen_type` field falls into one of the following five categories:

- `protein`
- `protein | protein`
- `protein | protein | protein`
- `protein | protein | protein | protein`
- `protein | protein | protein | protein | protein`

Furthermore, we only consider antibodies that either contain both heavy and light chains or consist of a single heavy chain (*i.e.*, nanobody), with a resolution no worse than 4.5 Å. Following DiffAb [Luo et al., 2022], we also utilize the Chothia scheme [Chothia and Lesk, 1987] for numbering the CDRs. After obtaining antibody sequences through parsing SAbDab's Chothia-scheme numbered antibodies, we further validate the CDR positions using the AbNumber[3] toolkit, which is based on ANARCI [Dunbar and Deane, 2016]. Moreover, we filter out antibody sequences that can not be correctly identified by AbNumber. These problematic sequences are found to either have incorrect lengths, missing regions among FR1/CDR1/FR2/CDR2/FR3/CDR3/FR4, or notably long CDR3 regions (*e.g.* chain I in PDB ID: 4k3e, which is an antibody heavy chain). Note that DiffAb also filters out sequences with abnormally long CDR3 regions in their preprocessing.

## A.2  Deduplication

In the TSV format summary file provided by SAbDab, there may be multiple antibody-antigen complex samples that correspond to the same PDB structure. We closely examine the variable domain sequences of the antibody heavy chains from these samples derived from the same PDB structure. After parsing the PDB structures and using the AbNumber tool to calibrate the CDRs regions, we employ AbNumber's alignment tool for pairwise comparisons and obtain some observations. Based on these observations, we develop a deduplication algorithm for these samples. Below, we present several typical examples of comparisons found in this process, focusing only on the CDRs sequences. First, consider the following pair of examples:

- VH sequence of Chain `H` in PDB `1bj1`:

    **FR1** GYTFTNY **FR2** NTYTGE **FR3** YPHYYGSSHWYFDV **FR4**

- VH sequence of Chain `K` in PDB `1bj1`:

    **FR1** GYTFTNY **FR2** NTYTGE **FR3** YPHYYGSSHWYFDV **FR4**

Both samples are associated with the same PDB structure, and their CDRs sequences are completely identical. For such samples, we believe that retaining just one is sufficient. We choose to retain the one with the longer VH sequence, as it provides more complete information. Now, consider the next set of examples:

- VH sequence of Chain `J` in PDB `6tyl`:

---

[3] 🐙 https://github.com/prihoda/AbNumber

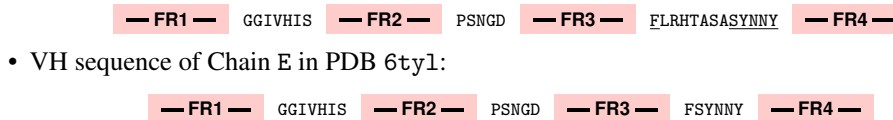

— FR1 —  GGIVHIS  — FR2 —  PSNGD  — FR3 —  FLRHTASASYNNY  — FR4 —

- VH sequence of Chain E in PDB `6tyl`:

— FR1 —  GGIVHIS  — FR2 —  PSNGD  — FR3 —  FSYNNY  — FR4 —

In the second set, the CDR-H1 and CDR-H2 regions are identical, but differences exist in the CDR-H3 sequence, where the lower sequence is noticeably shorter than the upper one. Upon closer inspection, it can be observed that the lower sequence lacks some amino acids present in the upper sequence. This situation may arise due to some unexpected uncertainties during the data collection process of the PDB structure. In our study, we regard these two sequences as essentially the same, and it is evident that the upper sequence provides more complete information, so we discard the lower sample. Now, consider the third set of examples:

- VH sequence of Chain K in PDB `7xj6`:

— FR1 —  GFTFSSS  — FR2 —  VVGSGN  — FR3 —  PNCNSTTCHDGFDI  — FR4 —

- VH sequence of Chain E in PDB `7xj6`:

— FR1 —  GGTFSSY  — FR2 —  IPILGI  — FR3 —  GTEYGDYDVSHD  — FR4 —

In the third set of examples, the sequences of CDR-H1 and CDR-H2 have the same length, but the amino acid types differ at certain positions. In the CDR-H3 sequence, although the lower one is shorter than the upper, this is not a case of missing parts as in the second example but rather indicates significant sequence differences. Thus, we consider these two chains as fundamentally different antibody sequences, and we retain both.

Based on these findings, we design the following algorithm to deduplicate the data obtained after processing in Section A.1. First, we use the Levenshtein distance to measure the differences between the CDRs sequences of two chains. The Levenshtein distance [Levenshtein, 1966] between two strings is defined as the minimum number of single-character edits (insertions, deletions, or substitutions) required to change one string into the other. In the examples above, the Levenshtein distance values for the three sets of sequences are 0, 7, and 20, respectively, illustrating varying degrees of similarity and difference.

We concatenate the CDRs sequences of each sample's VH sequence into a single string and then calculate the Levenshtein distance between these concatenated strings for each pair of samples when assessing differences. In our study, we set a threshold of 9 for the Levenshtein distance; if the distance does not exceed 9, the sequences are considered not significantly different and likely represent the same sequence, in which case one should be discarded. As some samples are nanobodies without a light chain, we only compare the CDRs of the VH sequence for all samples.

The algorithm is as follows: we group the samples by their associated PDB ID. For each group, we maintain a list to store the samples that are ultimately retained within the group. We first sort the samples in descending order based on the length of the VH sequence and add the longest sequence to the list, as it is considered to provide the most comprehensive information. Then, we iterate through the remaining sequences: if the Levenshtein distance to any sample in the list does not exceed the threshold, it is considered a duplicate; if the Levenshtein distance to all samples in the list exceeds the threshold, it is retained and added to the list. This process is detailed in Algorithm S1.

Following the processing detailed in Section A.1, we initially have 11,009 samples associated with 5,762 PDB structures. Upon completing the deduplication step, we retain 6,347 samples.

## A.3  Train & Validation & Test Sets Curation

As discussed in Section 4 of the main text, it is essential to consider the release date of the data to prevent data leakage when partitioning it into training, validation, and test sets.

We begin by applying MMSeqs2 [Steinegger and Söding, 2017] to cluster all antibodies based on their CDR-H3 sequences, using a 50% sequence identity threshold. These clusters form the basis for subsequent data partitioning and filtering.

Clusters that contain any structures with release dates prior to the cut-off date of September 30, 2021, are entirely included in the training set. This ensures that all pre-cut-off date data is used for training

---

**Algorithm S1** SAbDab Deduplication

---

    **Input:** a dataframe `DF` gathering samples sampled from Sec. A.1, distance_threshold;
1: all_reserved = []
    *# Group samples according to related `PDB_ID` and then iterate over each group*
2: **for** group **in** `DF`.groupby(‘`PDB_ID`’) **do**
        *# Sort the samples in descending order within current group based on the length of the VH sequence*
3:     sorted_group = group.sort_values(by=‘VH_SEQUENCE’, key=lambda x:x.str.len(), ascending=False)
        *# Reserve the sample with the longest VH sequence within the group*
4:     reserved=[sorted_group.iloc[0]]
        *# Iterate over every sample after the first one within the group*
5:     **for** entry **in** sorted_group.iloc[1:] **do**
6:         should_add_flag = True
7:         **for** reserved_entry **in** reserved **do**
8:           distance=levenshtein_distance(entry[‘VH_SEQUENCE’], reserved_entry[‘VH_SEQUENCE’])
9:           **if** distance $\leq$ distance_threshold **then**
10:             should_add_flag = False
11:             **break**
12:           **end if**
13:         **end for**
14:         **if** should_add_flag **then**
15:           reserved.append(entry)
16:         **end if**
17:     **end for**
        *# Merge reserved samples*
18:     all_reserved += reserved
19: **end for**
20: `filtered_DF` = DataFrame(all_reserved)
    **Output:** a dataframe `filtered_DF` composed of remaining data after Sec. A.2;

---

and avoids any risk of leakage into validation or test phases. For those clusters with data released on or after the cut-off date, we further divide them into validation and test sets to ensure these sets contain completely unseen data. The overall target ratio for the training, validation, and test clusters is approximately 9:0.5:0.5.

Due to Boltz-1's [Wohlwend et al., 2024] memory limitations, which do not support structures with more than 2000 residues/tokens on standard GPUs with 80GB memory (refer to this GitHub issue: https://github.com/jwohlwend/boltz/issues/17), we also filter out samples in the validation and test sets where the combined antibody-antigen complex exceeds 2000 residues.

Moreover, we incorporate two specific filters inspired by AlphaFold3 [Abramson et al., 2024] to improve data quality:

- Remove any sample where a complex contains a single chain with all residues marked as unknown.

- Exclude protein chains with consecutive $C_\alpha$ atoms that are more than 10 Åapart. This filter is applied only to the antibody chains and not to the antigen chains.

As a result of this process, we obtained 5,843 training samples from 2,570 clusters, 187 validation samples from 142 clusters, and 204 test samples from 144 clusters.

## A.4 MSA Construction

The Boltz-1 repository has already provided a way to obtain MSA results through the Colab-Fold [Mirdita et al., 2022] API. However, with limited computational resources, such a public remote server needs to queue incoming requests when handling multiple concurrent users, resulting in unstable processing speeds. To address this issue, we choose to build our own local server that exclusively processes our requests. Consistent with Boltz-1, we adopt these two databases for MSA search: `uniref30_2302`, `colabfold_envdb_202108`. After the setup, we can simply replace the default `msa_server_url` parameter in Boltz-1 with our local server URL to obtain paired and unpaired MSA results from the local MSA server. The server is built following the local setup guide provided in the ColabFold repository, available at: https://github.com/sokrypton/ColabFold/tree/main/MsaServer.

**Filtering.** We have already discussed in the main text the need to filter MSAs to prevent data leakage during both the training and inference stages. The filtering is done as follows: The MSA results provide an amino acid-by-amino acid alignment of the retrieved sequences with the query sequence. For each VH or VL query sequence, we examine each sequence retrieved one by one. We extract the segments that correspond to the three CDRs of the query sequence. If any of these segments have a sequence identity greater than or equal to a specified threshold with the corresponding CDR sequence, we remove the entire sequence from the final MSA results. In both the training and inference stages discussed in the main text, this threshold is set to 0.2.

## B  Supplementary Details about the Our Method

Due to space limitations in the main text, this section presents supplementray details on the methodological aspects of MFDesign. For implementation specifics, we defer to Appendix D.2.

### B.1  Token Cropping

As discussed in the Sec. 3.3, our model training consists of four stages, involving three token cropping strategies. The first strategy, employed in the first stage, is straightforward, involving inputting only the VH and VL tokens of the antibodies.

The other strategies are used in later stages. One of them, used specifically in the fourth stage, is a random token sampling method. For this, we directly adopt the cropping technique from the Boltz-1 [Wohlwend et al., 2024] paper, specifically utilizing the approach detailed in their Algorithm 2. This cropping method effectively handles complex of varying size by interpolating between *spatial* and *contiguous* cropping strategies. It defines "neighborhoods" around specific tokens, which are incrementally added based on their distance from a randomly selected crop center. By varying the size of neighborhoods, the algorithm can seamlessly transition from *spatial* to *contiguous* cropping.

The last strategy, used in the second, third, and fourth stages, focuses on cropping tokens around the antibody. This method, based on the above random token sampling method (Algorithm 2 from the Boltz-1 paper), is formalized in Algorithm S2 of our work, where we provide a comprehensive description of its tailored implementation for our specific need. In brief, this cropping approach first includes the VH and VL sequences of the antibody. Then, it randomly selects a token from the six CDRs as the center token. Tokens located at the antigen epitopes are then sorted in ascending order based on their distance to this selected center token. In this work, we define a residue on the antigen surface as an epitope residue if any internal atom is within a distance of less than a threshold of 10 Åfrom the atoms of the antibody CDRs. The remaining process follows Algorithm 2 from Boltz-1.

### B.2  Training Objective

In this work, we aim to tackle the task of co-designing antibody sequences and structures. For this purpose, we define two separate loss functions: one for the sequence diffusion training and another for the structure diffusion training. The structure diffusion loss, $\mathcal{L}_{\text{structure}}$, is directly taken from AF3, as defined by Equation 6 in AF3's supplementary materials. The sequence diffusion loss function is denoted as $\mathcal{L}_{\text{sequence}}$. Our final loss $\mathcal{L}$ is simply the sum of these two components:

$$\mathcal{L} = \mathcal{L}_{\text{sequence}} + \mathcal{L}_{\text{structure}},$$

Next, we give the detailed definition of $\mathcal{L}_{\text{sequence}}$ for the version using discrete diffusion for sequence. We adopt the D3PM-absorbing [Austin et al., 2021a], a discrete diffusion framework, for our sequence diffusion here. We denote the clean sample as $\{\mathbf{k}_i\}_0$ and we perform sequence diffusion only the CDRs within $\{\mathbf{k}_i\}_0$. The forward diffusion process $p\left(\{\mathbf{k}_i\}_t \mid \{\mathbf{k}_i\}_0\right)$ gradually corrups CDRs tokens by replacing them with special token <X> at each timestep $t$, via a discrete transition matrix that defines the probability of mutating a residue into <X>. The prior distribution at $t = T$ is a sequence where residues within CDRs are all <X> tokens. Our denoising model, **TokenDenoiser**, parameterized by $\theta$, $p_\theta\left(\{\hat{\mathbf{k}}_i\}_0 \mid \{\mathbf{k}_i\}_t, t\right)$, is trained to recover the original sequence from the absorbing version $\{\mathbf{k}_i\}_t$. To optimize the parameter set $\theta$, we maximize the likelihood of the denoising process on ground-truth tokens:

**Algorithm S2** Cropping Strategy Adopted in Our 2nd & 3rd & 4th Training Phases

---

**Input:** tokens, max token number $N_{\text{token}}^{\max}$, max atom number $N_{\text{atom}}^{\max}$, **neighborhood_sizes** = [0, 2, 4, . . ., 40];

1: **cropped_tokens** = []
2: Add all tokens in VH and VL sequence of the antibody to **cropped_tokens**
3: Sample **neighborhood_size** uniformly at random from **neighborhood_sizes**
4: Sample **center_token** uniformly within the tokens in the 6 CDRs of VH and VL
5: **sorted_epitope_tokens** ← sort tokens in the epitope of antigen by ascending their distance to **center_token**
6: **for token in sorted_epitope_tokens do**
7:     Let **chain_tokens** be the entries in the same chain as **token** (including **token**)
8:     **if** len(**chain_tokens**) ≤ **neighborhood_size then**
9:         **selected_tokens** = **chain_tokens**
10:     **else**
11:         **selected_tokens** = [**token**]
12:         **min_idx** ← index of **token** in **chain_tokens**
13:         **max_idx** ← index of **token** in **chain_tokens**
14:         **while** len(**selected_tokens**) < **neighborhood_size do**
15:             **min_idx** = **min_idx** - 1
16:             **max_idx** = **max_idx** + 1
17:             Let **selected_tokens** be the entries ∈ **chain_tokens** whose index in **chain_tokens** ∈ [**min_idx**, **max_idx**]
18:         **end while**
19:     **end if**
20:     Let **new_tokens** be the entries in **selected_tokens** that are not present in **cropped_tokens**
21:     **if** adding **new_tokens** to **cropped_tokens** would exceed $N_{\text{token}}^{\max}$ or max atom number $N_{\text{atom}}^{\max}$ **then**
22:         Break the **for** loop
23:     **else**
24:         Add **new_tokens** to **cropped_tokens**
25:     **end if**
26: **end for**
**Output: cropped_tokens**;

---

$$\mathcal{L}_{\text{sequence}} = \mathbb{E}_{\{\mathbf{k}_i\}_0, t} \left[ -\log p_\theta \left( \{\mathbf{k}_i\}_0 \mid \{\mathbf{k}_i\}_t \right) \right], \text{where } \{\mathbf{k}_i\}_t \sim p \left( \{\mathbf{k}_i\}_t \mid \{\mathbf{k}_i\}_0 \right).$$

## B.3 Sampling

In this paper, two sampling methods are employed: one for the antibody sequence and structure co-design task, and the other for structure prediction. We present our two sampling algorithms for the method employing discrete diffusion for sequence in this section.

### B.3.1 Sampling w/ *Replacement* Technique

The structure follows the sampling method from AF3 [Abramson et al., 2024] (refer to Algorithm 18 in AF3), which adopts the EDM [Karras et al., 2022] sampling method. For sequence sampling, we use the standard D3PM-absorbing [Austin et al., 2021a] sampling method.

In particular, when designing antibody CDRs, experimentally resolved co-crystal structures are provided, where all non-CDR regions (antibody framework sequences/structures, antigen sequences/structures, and the antibody-antigen binding pose) are given. This means our sampling must be conditioned on these fixed components.

In this work, we utilize a *replacement* sampling technique to allow the design of our CDRs to be conditioned on the given co-crystal structures. This technique is initially proposed and widely adopted in the image inpainting task [Lugmayr et al., 2022]. It has also been used in molecule and protein generation tasks [Schneuing et al., 2024, Trippe et al., 2023] to condition the sampling on given motifs. In simple terms, it involves replacing the generated contents corresponding to the fixed parts with their forward noised counterparts. In our sampling, we need to align the given ground-truth co-crystal structure to the denoised structure and then perform *replacement* operation.

Assuming the input has already been processed by the conditioning trunk to yield $\{\mathbf{s}_i^{\text{inputs}}\}, \{\mathbf{s}_i^{\text{trunk}}\}, \{\mathbf{z}_{ij}^{\text{trunk}}\}$, we then focus on the subsequent sampling process. We formalize our

**Algorithm S3** Sampling with *Replacement* Method

---

**Input:** raw inputs $\{\mathbf{f}^*\}$, trunk's outputs $\left\{\{\mathbf{s}_i^{\text{inputs}}\}, \{\mathbf{s}_i^{\text{trunk}}\}, \{\mathbf{z}_{ij}^{\text{trunk}}\}\right\}$, given ground-truth co-crystal structure $\{\mathbf{x}_l^{\text{GT}}\}$,

      input sequence $\{\mathbf{k}_i^{\text{input}}\}$ with CDRs marked in `<X>`,

      hyper-parameters required by EDM [Karras et al., 2022] sampling method $\{\sigma_{\text{data}}, \sigma_{\max}, \sigma_{\min}, \rho, \gamma_0, \gamma_{\min}, \lambda, \eta\}$;

1: **for** $t \in [0, 1, 2, \ldots, 199]$ **do**

    *# Define noise schedule for structure sampling*

2:    $c_t = \sigma_{\text{data}} \cdot \left(\sigma_{\max}^{1/\rho} + \frac{t}{200-1} \cdot \left(\sigma_{\min}^{1/\rho} - \sigma_{\max}^{1/\rho}\right)\right)^{\rho}$

    *# Define noise schedule for sequence sampling*

3:    $m_t = 1 - \frac{t}{200-1}$

4: **end for**

    *# Initialize structure via sampling from a Gaussian noise*

5: $\mathbf{x}_l \sim c_0 \cdot \mathcal{N}\left(\vec{\mathbf{0}}, \mathbf{I}_3\right)$

    *# Input sequences naturally serve as initialization because CDRs are all already in the absorbing state*

6: $\{\mathbf{k}_i^{\text{noisy}}\} = \{\mathbf{k}_i^{\text{input}}\}$

7: **for** $t \in [1, 2, \ldots, 199]$ **do**

8:    $\{\mathbf{x}_l\} = \textbf{CentreRandomAugmentation}(\{\mathbf{x}_l\})$       ▶ Refer to Algorithm 19 in AF3

9:    $\gamma = \gamma_0$ **if** $c_t > \gamma_{\min}$ **else** 0

10:  $\sigma = c_{t-1}(\gamma + 1)$

11:  $\xi_l = \lambda\sqrt{\sigma_t^2 - c_{t-1}^2} \cdot \mathcal{N}\left(\vec{\mathbf{0}}, \mathbf{I}_3\right)$      Same as AF3, which follows EDM sampling method

12:  $\mathbf{x}_l^{\text{noisy}} = \mathbf{x}_l + \xi_l$

13:  $\{\mathbf{x}_l^{\text{denoised}}\}, \{\mathbf{k}_i^{\text{denoised}}\} = \textbf{CoDiffusionModule}\left(\{\mathbf{x}_l^{\text{noisy}}\}, \{\mathbf{k}_i^{\text{noisy}}\}, \sigma, \{\mathbf{f}^*\}, \{\mathbf{s}_i^{\text{inputs}}\}, \{\mathbf{s}_i^{\text{trunk}}\}, \{\mathbf{z}_{ij}^{\text{trunk}}\}\right)$   ▶ Alg. 1

    *# Replacement sampling technique (Lines 14-15) [Lugmayr et al., 2022, Schneuing et al., 2024]*

14:  $\mathbf{x}_l^{\text{GT-aligned}} = \textbf{RigidAlign}\left(\{\mathbf{x}_l^{\text{GT}}\}, \{\mathbf{x}_l^{\text{denoised}}\}\right)$       ▶ Refer to Algorithm 28 in AF3

15:  $\mathbf{x}_l^{\text{denoised}} = \mathbf{x}_l^{\text{denoised}}$ **if** atom $l$ is within CDRs **else** $\mathbf{x}_l^{\text{GT-aligned}}$

16:  $\delta_l = (\mathbf{x}_l - \mathbf{x}_l^{\text{denoised}})/\sigma$

17:  $dt = c_t - \sigma$      Same as AF3, which follows EDM sampling method

18:  $\mathbf{x}_l \leftarrow \mathbf{x}_l^{\text{noisy}} + \eta \cdot dt \cdot \delta_l$

    *# Randomly sample a probability from a uniform distribution between 0 and 1 for each token*

19:  $p_i \sim \mathcal{U}(0, 1)$

20:  $\mathbf{k}_i^{\text{denoised}} = \mathbf{k}_i^{\text{denoised}}$ **if** residue $i$ is within CDRs **else** $\mathbf{k}_i^{\text{input}}$      Follow D3PM-absorbing

21:  $\mathbf{k}_i^{\text{noisy}} = $ one-hot encoding of `<X>` **if** $p_i < m_t$ **and** residue $i$ is within CDRs **else** $\mathbf{k}_i^{\text{denoised}}$    sampling method

22: **end for**

23: $\mathbf{x}_l^{\text{GT-aligned}} = \textbf{RigidAlign}\left(\{\mathbf{x}_l^{\text{GT}}\}, \{\mathbf{x}_l\}\right)$       ▶ Refer to Algorithm 28 in AF3

24: $\mathbf{x}_l = \mathbf{x}_l$ **if** atom $l$ is within CDRs **else** $\mathbf{x}_l^{\text{GT-aligned}}$

25: $\{\mathbf{k}_i\} = \{\mathbf{k}_i^{\text{denoised}}\}$

   **Output:** predicted final sequence $\{\mathbf{k}_i\}$, predicted final structure $\{\mathbf{x}_l\}$;

---

sampling method for antibody sequence and structure co-design in Algorithm S3. In fact, the raw inputs $\{\mathbf{f}^*\}$ include $\{\mathbf{k}_i^{\text{input}}\}$, but we choose to highlight them separately.

### B.3.2 Sampling w/o *Replacement* Technique

For the structure prediction task, we sample from scratch. Therefore, we do not need to input a given structure as in Algorithm S3, nor do we use the *replacement* sampling technique. As a result, the sampling algorithm used here omits a given structure as input and also skips Lines 14-15 from Algorithm S3, which correspond to the *replacement* technique, while the rest remains consistent with Algorithm S3. If the input antibody sequence contains the `<X>` token, we will simultaneously predict its amino acid type while predicting the structure.

### B.4 Sequence Diffusion in the Continuous Space

In the main body of this paper, we present an implementation utilizing D3PM-absorbing [Austin et al., 2021a], a discrete diffusion method, for sequence diffusion. Additionally, we also implement a continuous version. Following the methodology of prior research [Hoogeboom et al., 2022], we introduce Gaussian noise directly to the one-hot encoding of residue types to handle discrete data through continuous diffusion.

As noted in Sec. 3.2, the noise level for AF3's structure diffusion training $\sigma$ is sampled from $\sigma_{\text{data}} \cdot \exp(-1.2 + 1.5 \cdot \mathcal{N}(0, 1))$. For inference, AF3 defines a sequence of decreasing noise levels: $\left\{\sigma_{\text{data}} \cdot \left(\sigma_{\max}^{1/\rho} + \frac{t}{200-1} \cdot \left(\sigma_{\min}^{1/\rho} - \sigma_{\max}^{1/\rho}\right)\right)^{\rho} \mid t = 0, 1, \ldots, 199\right\}$. $\sigma_{\text{data}}, \sigma_{\max}, \sigma_{\min}$ and $\rho$ are hyper-parameters. For structure diffusion, we adhere to the default settings in AF3 for these parameters.

Since our diffusion now is also based on continuous diffusion, we can also apply the EDM [Karras et al., 2022] training method to sequence diffusion. Similar to structure diffusion, we directly sample

the noise level instead of first sampling a time step and then using a predefined noise scheduler to obtain the corresponding noise level. By sharing the noise level between structure and sequence, we can easily align their training. We define a scaling factor $\omega$ and set it to $1/4$ in our experiments. In our implementation, we initially set the scaling factor to $1/4$. Empirical results show this value already delivers significantly better performance than baselines. Due to limited computing resources, we do not perform extensive hyperparameter tuning. The noise level for the structure is sampled from $\sigma_{\text{data}} \cdot \exp(-1.2 + 1.5 \cdot \mathcal{N}(0, 1))$, and then this sampled noise level is multiplied by $\omega$ to serve as the noise level for the sequence. This ensures that the noise levels for the sequences are proportional to those for the structure. Once the noise level for the structure is determined, the noise level for the sequence diffusion can be obtained using $\omega$, and noise is then added to the one-hot encodings of tokens to obtain $\{\mathbf{k}_i^{\text{noisy}}\}$. Diffusion is still applied only to the CDRs.

In this following, we present how to input the corresponding noisy sequence into the **CoDiffusion-Module** at each sampled timestep. During training, we add noise to the one-hot encodings of tokens to obtain $\{\mathbf{k}_i^{\text{noisy}}\}$. Similar to the method used in discrete sequence diffusion, we then replace the specified dimension in $\{\mathbf{s}_i^{\text{inputs}}\}$ with $\{\mathbf{k}_i^{\text{noisy}}\}$. For each denoising step in inference, we directly add noise to the token-level probability distributions, $\{\mathbf{h}_i\}$, which represents the probability distribution of each amino acid type for each token. This is obtained by inputting previously predicted $\{\mathbf{k}_i^{\text{denoised}}\}$ after last step into the SOFTMAX function. Then, we replace the specified dimensions in $\{\mathbf{s}_i^{\text{inputs}}\}$ with $\{\mathbf{h}_i\}$.

### B.5 Limitations and Future Work

One limitation of our current approach is that we only generate backbone atoms for CDRs and lack support for non-standard residues, requiring an additional side-chain packing step to obtain full-atom structures. Accordingly, we aim to develop an end-to-end model capable of direct all-atom structure prediction in the future. Additionally, we currently use full-parameter fine-tuning and plan to explore more efficient strategies to reduce computational costs while maintaining result quality.

On the other hand, a methodological limitation arises from our use of a discrete approach, which requires forcing continuous processes into discrete time steps for alignment. This discretization can impair model performance. A promising future direction is to adopt a discrete diffusion model capable of operating within a continuous time/schedule. This would allow sequence and structure generation to be perfectly synchronized within a unified, continuous training framework, enabling each component to operate in its optimal modality.

### B.6 Broader Impacts

In this work, we adapt AlphaFold3-like folding models with specific modifications to enable the co-design of antibody sequence and structure, in addition to their existing structure prediction capabilities. Our societal impact lies in potentially accelerating the development of new antibodies, which could improve healthcare by providing more efficient treatments. So far, we have not identified any significant negative impacts, but we remain attentive to any potential concerns that may arise.

## C  Details about the Adopted Metrics

In this section, we provide more details about the metrics we used in this paper for evaluation.

### C.1  Metrics Adopted in Sec. 4.1

**AAR.** It measures the accuracy of the generated CDRs sequences by comparing the amino acid identity with native sequences. It calculates the percentage of amino acids in the generated sequence that match the reference sequence.

**RMSD.** RMSD measures the deviation of the $C_\alpha$ atom coordinates between generated and native CDRs. Following DiffAb [Luo et al., 2022], the antibody frameworks are aligned prior to RMSD calculation. In this work, this alignment is specifically based on the $C_\alpha$ atoms.

**Loop-AAR.** This metric evaluates the accuracy of amino acid recovery specifically for the central loop residues of CDR-H3, the region most critical for antigen binding. Consistent with [Zhu et al.,

2024], we exclude specific residues from our analysis. For the Chothia numbering scheme [Chothia and Lesk, 1987], we exclude the initial and terminal two residues of CDR-H3. In contrast, for the IMGT scheme [Lefranc et al., 2003], we exclude the first four and the last two residues.

**Loop-RMSD.** This metric assesses the structural deviation of the central loop residues within CDR-H3 between generated and native structures, focusing on the region critical for antigen binding. Consistent with Loop-AAR, we follow the loop region definitions specific to each numbering scheme. Similar to the RMSD calculation, the antibody frameworks are aligned before computing Loop-RMSD.

**IMP.** The Improvement Percentage (IMP) is determined by evaluating the binding energies of antibody-antigen complexes, calculated using the *InterfaceAnalyzer* with the *ref2015* score function in PyRosetta toolkit [Chaudhury et al., 2010]. IMP represents the fraction of designed complexes that achieve lower binding energies compared to their natural counterparts.

Metrics calculated from structural data are computed after we perform side-chain packing and relaxation.

## C.2 Metrics Adopted in Sec. 4.2

**RMSD.** Here, RMSD measures the root-mean-square deviation of all $C_\alpha$ atom coordinates between generated and native protein structures. For this calculation, the entire protein structure is aligned, focusing on $C_\alpha$ atoms to provide an overall measure of structural accuracy.

**Loop-RMSD.** It evaluates the structural deviation of $C_\alpha$ atoms specifically within the central loop of the CDR-H3 region. This metric is calculated after aligning the entire protein structure and focuses on prediction accuracy in this highly variable and critical region for antigen binding.

**TM-score.** TM-score assesses the similarity between predicted and native protein structures, with scores ranging from 0 to 1. A higher TM-score indicates better global similarity and is particularly sensitive to the overall fold of the protein, rather than local variations.

Given that these metrics only require backbone predictions, and as noted by AlphaFold 3 that that the relax postprocessing step is rarely needed when using AF3, we do not perform side-chain packing or relaxation here. The models' prediction results are used directly for evaluation.

# D  More Details about the Implementations

Due to our use of different data splits from those in the original papers of the baselines, it is necessary to retrain these baseline models to ensure a fair comparison. In this section, we provide detailed information on the implementation of both the baseline models and MFDesign. All experiments run on a single node consisting of $8 \times$ H100 GPUs with 80 GB HBM3 each (aggregated GPU memory of 640 GB), $2 \times$ Intel Xeon Platinum 8468 processors comprised of 48 CPUs each (total 96 cores, 192 threads).

## D.1 Details about the Implementations of Baselines

**DiffAb**[4] [Luo et al., 2022] We implement the baseline using the code provided by the authors, and we adhered to the hyper-parameter settings detailed in the *codesign_multicdrs.yml* file from the repository for our training configuration. DiffAb employs the Chothia numbering scheme, which is consistent with our approach.

**dyMEAN**[5] [Kong et al., 2023a] We utilize the code provided by the authors and set our training parameters according to the hyper-parameters specified in the *multi_cdr_design.json* file in the repository. dyMEAN employs the IMGT scheme [Lefranc et al., 2003] for numbering antibodies. We choose to adhere to its original selection regarding the scheme as Zhu et al. do.

**Remark.** The official dyMEAN code does not support nanobody processing, which results in a reduced number of samples for evaluation in dyMEAN as compared to the full test set. In contrast,

---

[4] https://github.com/luost26/diffab
[5] https://github.com/THUNLP-MT/dyMEAN

both DiffAb and our method are capable of effectively handling nanobodies, allowing for evaluation on the complete test set.

## D.2 Details about the Implementations of Our Method

Since AlphaFold3 has not open-sourced its training scripts, in this work, we base our approach on Boltz-1, a reproduced version of AF3 by Wohlwend et al., which provides these scripts. Additionally, Boltz-1 maintains consistency with AlphaFold3 in terms of variable and function definitions, facilitating a clearer understanding of the implementation alongside the original AlphaFold3 paper.

Our primary modification compared to AF3 is the addition of a denoising network for sequences, which we implement using a simple MLP composed of several LINEAR layers connected by GELU activation functions. We observe that in the implementation of DiffAb [Luo et al., 2022], an additional feature is encoded for each amino acid to indicate its region. We adopt this approach in our implementation and find it beneficial for accelerating early convergence of the model. Specifically, we include a feature indicating which chain a residue belongs to, specifying whether it is the heavy chain, light chain, or antigen. For antibody sequences (VH and VL), another feature specifies the region within the chain, such as FR1, CDR1, FR2, CDR2, FR3, CDR3, or FR4. For antigen residues in the antibody co-design task, where the baseline provides epitope information, we include a feature to identify if an antigen residue is part of the epitope region. However, in complex structure prediction, we do not differentiate based on epitopes, as incorporating such information does not align with the bind-docking setting. These additional features, along with the previously obtained token-level representations, are fed into the sequence denoising network.

For our models used in the co-design of antibody sequences and structures, the implementation of both discrete and continuous diffusion for sequence follows the same training approach:

- **1st-stage:** A warm-up strategy is employed, where the learning rate increases linearly from $0$ to a predefined value $5 \times 10^{-4}$ over 10 steps. Subsequent to this warm-up phase, the learning rate decays by a factor of $0.999$ every 100 steps. The max token number is set to 256. The batch size for a single GPU is 6, thus amounting to 48 samples per training step. This stage consists of a total of $5,000$ training steps.

- **2nd-stage:** Training continues for a total of $2,000$ steps, starting with an initial learning rate that is set directly. The learning rate then decays by a factor of $0.999$ every 100 steps. The max token number is set to 384, and the batch size for a single GPU is set to 3, resulting in 24 samples per training step.

- **3rd-stage:** Training continues for a total of $5,000$ steps, starting with an initial learning rate that is set directly. The learning rate then decays by a factor of $0.999$ every 100 steps. The max token number is set to 512, and the batch size for a single GPU is set to 1, resulting in 8 samples per training step.

- **4th-stage:** Training continues for a total of $9,000$ steps, starting with an initial learning rate that is set directly. The learning rate then decays by a factor of $0.999$ every 100 steps. The max token number is set to 512, and the batch size for a single GPU is set to 1, resulting in 8 samples per training step.

# E More Experimental Results

## E.1 Results of Generating One CDR at A Time

Some prior works [Luo et al., 2022] have adopted single-CDR design settings, where individual CDRs are designed independently while treating other regions as fixed structural contexts. While this approach simplifies computational complexity, compared to our simultaneous design of all six CDRs at once, such setting presents two fundamental limitations for real-world antibody development:

- The setting of designing only one CDR at a time actually simplifies the real problem. In this simplified setting, for example when we want to design CDR-H3, the other CDRs are directly given as ground-truth context, so a separate model needs to be trained for the design of each corresponding CDR region. However, antibody-antigen binding relies on the cooperative action of

Table S1: Comparison of designed CDR-H3 for typical antibodies in terms of AAR and RMSD metrics under two settings, with the **best** and second-best results highlighted. The comparison evaluates CDR-H3 generation under two approaches: (1) separate CDR-H3 generation versus (2) simultaneous generation with other CDRs. Due to limited computational resources, we only trained the discrete version of our model for this analysis.

| | | Separately Design | | | | Simultaneously Design | |
|---|---|---|---|---|---|---|---|
| CDR | Method | AAR / % ↑ | RMSD / Å ↓ | CDR | Method | AAR / % ↑ | RMSD / Å ↓ |
| H3 | **DiffAb** | 24.41 | 3.89 | H3 | **DiffAb** | 22.26 | 4.29 |
| | **dyMEAN** | 39.03 | 3.82 | | **dyMEAN** | 34.69 | 6.15 |
| | **Ours** | **67.79** | **3.44** | | **Ours** | **65.04** | **3.71** |

all six CDRs. In real-world scenarios, when designing antibodies against an unseen antigen, all six CDR regions need to be designed, where no ground-truth exists for any CDR.

- Furthermore, if each CDR is generated separately one by one, there would be an order dependency. It's difficult to determine in what order the six CDRs should be generated sequentially.

In contrast, simultaneously generating all six CDRs is more practically valuable for real-world applications.

But we still supplement the results for single-CDR design in this study. Due to limited time and computational resources, we only train one CDR-H3-specific design model for both our method and baselines. We specifically select this CDR because it plays the primary role in antibody-antigen binding. The results are presented in Table S1. We can see that all methods perform better when generating CDR-H3 alone compared to jointly generating CDR-H3 with other CDRs, indicating that the joint-generation setting is more challenging, as it requires simultaneously modeling the structural and sequence distributions of all six CDR regions, which is more complex. Particularly, our method outperforms baselines by a notable margin in both scenarios (CDR-H3-only and full six-CDR generation) across all metrics.

### E.2 Comparison against More Baselines: RFantibody and IgGM

In this study, we provide comparison results against two additional baseline methods: **RFantibody** [Bennett et al., 2024] and **IgGM** [Wang et al., 2025]. We exclude them from our main analysis for the following reasons:

- **RFantibody.** We would like to humbly point out that RFantibody follows a sequential two-stage approach: initially generating structures using RFDiffusion [Watson et al., 2023], followed by sequence generation with ProteinMPNN [Dauparas et al., 2022]. This differs fundamentally from our co-design setting. Therefore, since its setting differs from ours and does not fall under the scope of antibody sequence and structure co-design studied in this paper, we do not compare this work in the main paper.

- **IgGM.** IgGM was accepted at ICLR 2025 (conference date: late April), making it essentially contemporaneous with our NeurIPS 2025 submission deadline. Furthermore, the authors only release the inference code and model weights. The critical training framework remains unavailable. This partial disclosure prevents scientifically rigorous evaluation, as we cannot retrain the model on our standardized data splits.

Despite these above constraints, we include supplemental comparisons against them and summarize the results in Table S2, Table S3 and Table S4. The results still demonstrates superior performance of our method, further validating its advantages.

### E.3 Metric beyond the AAR and RMSD: Binding Energy $\Delta G$

The evaluation metrics employed in our main text - AAR and RMSD, are standard evaluation metrics in machine learning-based antibody design literature [Luo et al., 2022, Kong et al., 2023a, Martinkus et al., 2024]. In Sec. 4, we have reported results using the IMP metric, another energy-related evaluation indicator, where we outperform baselines by a notable margin.

Table S2: Comparison of simultaneously designed CDRs for typical antibodies in terms of AAR and RMSD metrics, with the **best** and the second-best highlighted for each CDR. We additionally compared our results with RFantibody and IgGM. Since neither method has released their training code publicly, we directly used the checkpoints provided by the authors for testing. It should be noted that these comparisons are not entirely fair because the models were not trained on the same dataset as ours. Therefore, these results should only be considered as providing limited reference value. In particular, IgGM was just accepted at ICLR 2025, making it essentially contemporaneous with our NeurIPS submission deadline. ***Findings:*** The results show our method maintains leading performance, outperforming all baselines in both metrics except for CDR-L3's RMSD where IgGM performs better. Specifically, for CDR-H3, which is the key region determining antibody-antigen binding, our method achieves substantially better AAR and RMSD results than all baseline methods by a notable margin.

| CDR | Method | AAR / % ↑ | RMSD / Å ↓ | CDR | Method | AAR / % ↑ | RMSD / Å ↓ |
|---|---|---|---|---|---|---|---|
| H1 | DiffAb | 60.92 | 1.52 | L1 | DiffAb | 56.72 | 1.43 |
|  | dyMEAN | 70.31 | 1.65 |  | dyMEAN | 73.06 | 1.58 |
|  | RFantibody | 27.08 | 3.01 |  | RFantibody | 21.82 | 3.23 |
|  | IgGM | 73.01 | 1.70 |  | IgGM | 74.02 | 1.44 |
|  | MFDesign-C | 74.89 | 1.48 |  | MFDesign-C | 82.98 | 1.41 |
|  | MFDesign-D | 74.95 | 1.61 |  | MFDesign-D | 82.98 | 1.65 |
| H2 | DiffAb | 33.53 | 1.44 | L2 | DiffAb | 55.08 | 1.21 |
|  | dyMEAN | 66.38 | 1.47 |  | dyMEAN | 76.04 | 1.23 |
|  | RFantibody | 22.23 | 3.03 |  | RFantibody | 21.66 | 2.19 |
|  | IgGM | 63.89 | 1.73 |  | IgGM | 73.11 | 1.26 |
|  | MFDesign-C | 65.59 | 1.26 |  | MFDesign-C | 88.84 | 1.00 |
|  | MFDesign-D | 67.54 | 1.44 |  | MFDesign-D | 87.81 | 1.15 |
| H3 | DiffAb | 22.26 | 4.29 | L3 | DiffAb | 43.03 | 1.80 |
|  | dyMEAN | 34.69 | 6.15 |  | dyMEAN | 52.69 | 1.59 |
|  | RFantibody | 8.07 | 6.01 |  | RFantibody | 14.63 | 3.33 |
|  | IgGM | 32.45 | 4.07 |  | IgGM | 61.02 | 1.51 |
|  | MFDesign-C | 63.13 | 3.54 |  | MFDesign-C | 78.93 | 1.55 |
|  | MFDesign-D | 65.04 | 3.71 |  | MFDesign-D | 80.15 | 1.69 |

Table S3: Results on typical antibodies with simultaneously designed CDRs, highlighting the **best** and the second-best. We additionally compared our results with RFantibody and IgGM. Since neither method has released their training code publicly, we directly used the checkpoints provided by the authors for testing. It should be noted that these comparisons are not entirely fair because the models were not trained on the same dataset as ours. Therefore, these results should only be considered as providing limited reference value. In particular, IgGM was just accepted at ICLR 2025, making it essentially contemporaneous with our NeurIPS submission deadline. ***Findings:*** The results show that RFantibody performs slightly better than our method on the IMP metric, while our method significantly surpasses all baselines on other metrics. Recall that in Table S2, although RFantibody achieves better IMP scores, it performs poorly on both AAR and RMSD metrics - these two being the standard evaluation metrics in machine learning-based antibody design literature. We would like to note again that RFantibody does not fall under the sequence-structure co-design methods that are the focus of our paper.

| Metric | DiffAb | dyMEAN | RFantibody | IgGM | MFDesign-C | MFDesign-D |
|---|---|---|---|---|---|---|
| **Loop-AAR / % ↑** | 15.86 | 20.80 | 9.69 | 25.41 | 61.71 | 63.38 |
| **Loop-RMSD / Å ↓** | 5.03 | 7.84 | 6.94 | 4.68 | 4.13 | 4.28 |
| **IMP / % ↑** | 53.35 | 5.60 | 68.07 | 45.34 | 65.37 | 59.16 |

In this section, we have included additional results on another energy-related metric, *i.e.* binding energy $\Delta G$. The related results are summarized in Table S5.

## E.4 Diversity Analysis

While prior antibody co-design work (including DiffAb) omitted sequence diversity evaluation, we argue that both quality and diversity metrics are crucial. The tradeoff between the quality and diversity of generated samples can be manipulated by adjusting the temperature parameter during sampling. We design the following protocol to evaluate the diversity:

- We fist implements a temperature parameter ($\{0.1, 0.5, 1\}$) to scale sampling logits.
- We then define a diversity metric with its details as follows:

Table S4: Comparison of generated nanobodies with simultaneously designed CDRs, assessed based on AAR and RMSD for CDR-H1, H2, and H3, as well as Loop-AAR, Loop-RMSD, IMP metrics, with the **best** and the second-best results highlighted. Note that we do not compare against dyMEAN here, as its source code does not support processing nanobodies. We additionally compared our results with RFantibody and IgGM. Since neither method has released their training code publicly, we directly used the checkpoints provided by the authors for testing. It should be noted that these comparisons are not entirely fair because the models were not trained on the same dataset as ours. Therefore, these results should only be considered as providing limited reference value. In particular, IgGM was just accepted at ICLR 2025, making it essentially contemporaneous with our NeurIPS submission deadline. ***Findings:*** The results clearly show that our method maintains superior overall performance. The only exceptions are marginally lower performance on CDR-H2's RMSD compared to DiffAb. Crucially, for the most challenging CDR-H3 region, which is the key determinant of antibody-antigen binding, our method achieves significantly better results than all baselines in both AAR and RMSD metrics.

| Method | CDR-H1 | | CDR-H2 | | CDR-H3 | | Loop-AAR / % | Loop-RMSD / Å | IMP / % |
|---|---|---|---|---|---|---|---|---|---|
| | AAR / % | RMSD / Å | AAR / % | RMSD / Å | AAR / % | RMSD / Å | | | |
| DiffAb | 45.17 | 2.52 | 31.11 | **1.54** | 15.39 | 4.71 | 10.51 | 5.50 | 30.93 |
| RFantibody | 22.65 | 2.52 | 19.67 | 1.91 | 9.12 | 4.96 | 10.05 | 5.68 | 43.02 |
| IgGM | 58.08 | 2.86 | 40.19 | 2.22 | 17.84 | 4.77 | 11.43 | 5.49 | 25.58 |
| MFDesign-C | 65.71 | **2.48** | 55.60 | 1.62 | 61.40 | **3.71** | **61.01** | **4.26** | 43.14 |
| MFDesign-D | **67.34** | 2.62 | **58.93** | 1.78 | **63.51** | 3.84 | 60.80 | 4.41 | 38.26 |

Table S5: Results on typical antibodies with simultaneously designed CDRs in terms of binding energy $\Delta G$ (kcal/mol, $\downarrow$). *Reference* corresponds the results for the ground-truths. The **best** and the second-best are highlighted respectively. We additionally compared our results with RFantibody and IgGM. Since neither method has released their training code publicly, we directly used the checkpoints provided by the authors for testing. It should be noted that these comparisons are not entirely fair because the models were not trained on the same dataset as ours. Therefore, these results should only be considered as providing limited reference value. In particular, IgGM was just accepted at ICLR 2025, making it essentially contemporaneous with our NeurIPS submission deadline. ***Findings:*** Our experimental results demonstrate that the antibodies designed by our method achieve significantly better average binding energy compared to other baseline methods. In terms of best binding energy, only RFantibody shows marginally better performance than ours. We would like to emphasize that RFantibody does not belong to the sequence-structure co-design paradigm that is the primary focus of this study, representing a fundamentally different technical approach.

| Method | $\Delta G \downarrow$ | |
|---|---|---|
| | best | avg. |
| *Reference* | -60.08 | -6.60 |
| **DiffAb** | -64.51 | 16.97 |
| **dyMEAN** | -13.10 | 3542.55 |
| **RFantibody** | **-77.26** | 2.44 |
| **IgGM** | -52.81 | -5.04 |
| **MFDesign-C** | -67.98 | **-21.72** |
| **MFDesign-D** | -62.47 | -20.83 |

- We compute pairwise CDR sequence identity across candidates.

- We average the identity scores per sample, then compute the mean across the entire test set.

- We use 1 - score (since lower original values indicate better diversity) as final metric.

The results are demonstrated in Fig. S1. We can see that at temperature = 1 (no logits scaling), our method achieves significantly higher AAR than DiffAb while showing lower diversity. We maintain that quality outweighs diversity, as highly diverse but low-quality samples would be biologically meaningless. At temperature = 0.1, our method achieves comparable diversity to DiffAb, while delivering substantially better overall AAR (12.59% vs DiffAb's 5.68%) - more than a two-fold improvement, clearly demonstrating our method's superior generation quality.

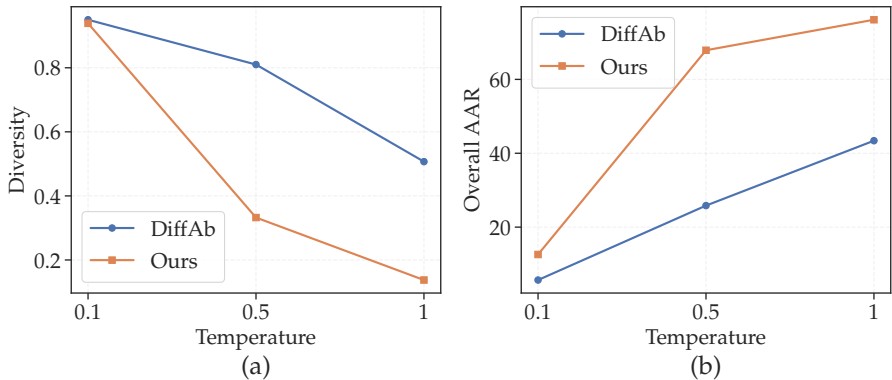

Figure S1: **Figure: (a)** Comparison of diversity variation with temperature; **(b)** Comparison of overall AAR variation with temperature.

Table S6: Performance comparison on the structure prediction task, evaluated by RMSD, Loop-RMSD, and TM-score.

| Given CDR? | Fine-tuned? | H1 | H2 | H3 | L1 | L2 | L3 |
|:---:|:---:|:---:|:---:|:---:|:---:|:---:|:---:|
| ✗ | ✗ | 3.72 | 2.21 | 4.79 | 3.55 | 1.51 | 2.83 |
| ✗ | ✓ | 2.01 | 1.79 | 4.22 | 1.54 | 1.32 | 1.59 |
| ✓ | ✗ | 1.64 | 1.40 | 3.04 | 1.12 | 1.08 | 1.16 |
| ✓ | ✓ | 1.91 | 1.70 | 3.91 | 1.38 | 1.23 | 1.39 |

Table S7: Comparison of antigen structure prediction performance, with and without the fourth training stage, evaluated using RMSD and TM-score, with **best results** highlighted.

| w/ 4-th training stage? | RMSD / Å ↓ | TM-score / % ↑ |
|:---:|:---:|:---:|
| ✗ | 9.87 | 74.91 |
| ✓ | **9.59** | **76.72** |

## E.5 Per-CDR Structural Accuracy Analysis

To complement the integrated RMSD metrics in Table 4, we conduct additional per-CDR evaluations and summarize the results in Table S6. It can be observed that the RMSD values for individual CDRs are consistent with the overall RMSD results presented in our main text: when amino acid types of CDRs are not provided in the input sequence, our fine-tuned Boltz-1 performs significantly better than the original model; when CDR sequences are provided, the fine-tuned version performs slightly worse than the original model, but still achieves good results. We have already provided explanations for both observations in Section 4.2.

## E.6 Ablation of the 4th Training Stage

Recall that we adopt a four-stage training approach. The first three stages primarily focus on selecting tokens from regions around the antibody and its nearby areas, which could result in inadequate training for antigen regions far from the antibody. To address this limitation, we introduce a fourth training stage specifically designed to ensure adequate learning of these distant antigen regions.

Here, we use our model to predict the structures of antigens, comparing the performance with and without the fourth stage of training. For each antigen in the test set from Sec. 4.1, we generate 5 predictions. We employ RMSD and TM-score as evaluation metrics and summarize the results in Table S7. We use the MF-Design with discrete sequence diffusion for this study. We find that, after undergoing the fourth training stage, the model shows improvements in both metrics compared to the model trained with only the first three stages.

Table S8: Antibody design results on the RAbD dataset (mean±std) over all CDRs. Higher AAR and lower RMSD are better. Best and second-best are highlighted.

| CDR | Method | AAR / % ↑ | RMSD / Å ↓ | CDR | Method | AAR / % ↑ | RMSD / Å ↓ |
|---|---|---|---|---|---|---|---|
| H1 | DiffAb | 67.03±2.25 | 1.10±0.07 | L1 | DiffAb | 57.95±1.75 | 0.96±0.04 |
| | dyMEAN | 79.15±2.32 | **1.03±0.08** | | dyMEAN | 77.00±1.81 | 0.97±0.05 |
| | MFDesign-C | 79.06±1.16 | 1.08±0.06 | | MFDesign-C | **86.80±1.38** | **0.87±0.04** |
| | MFDesign-D | **79.55±1.13** | 1.15±0.07 | | MFDesign-D | 86.31±1.52 | 1.12±0.05 |
| H2 | DiffAb | 35.16±1.36 | 1.13±0.06 | L2 | DiffAb | 55.84±1.80 | 0.92±0.04 |
| | dyMEAN | **73.87±2.82** | 0.97±0.05 | | dyMEAN | 88.30±3.28 | 0.79±0.04 |
| | MFDesign-C | 70.93±1.91 | **0.89±0.04** | | MFDesign-C | **92.85±1.20** | **0.68±0.02** |
| | MFDesign-D | 72.31±2.01 | 1.04±0.04 | | MFDesign-D | 90.46±1.39 | 0.80±0.02 |
| H3 | DiffAb | 26.66±1.30 | 2.88±0.17 | L3 | DiffAb | 45.15±1.35 | 1.21±0.07 |
| | dyMEAN | 38.46±2.01 | 4.44±0.20 | | dyMEAN | 57.88±2.70 | 1.20±0.09 |
| | MFDesign-C | 77.87±1.93 | **2.26±0.13** | | MFDesign-C | **86.17±1.47** | **1.12±0.07** |
| | MFDesign-D | **78.61±1.69** | 2.35±0.12 | | MFDesign-D | 85.26±1.73 | 1.24±0.07 |

Table S9: Performance on loop regions and overall improvement (IMP) on the RAbD dataset (mean±std). Higher AAR/IMP and lower RMSD are better.

| Method | Loop-AAR / % ↑ | Loop-RMSD / Å ↓ | IMP / % ↑ |
|---|---|---|---|
| DiffAb | 20.21±1.32 | 3.44±0.19 | 22.29±5.42 |
| dyMEAN | 24.28±2.68 | 6.00±0.24 | 0.00±0.00 |
| MFDesign-C | 73.76±2.82 | **2.69±0.15** | **48.14±0.07** |
| MFDesign-D | **75.27±2.43** | 2.75±0.14 | 41.95±6.42 |

## E.7 Results on RAbD Dataset

To further assess the generalizability of our method, we conducted additional experiments on the RAbD benchmark, a widely used dataset for antibody sequence and structure co-design. While our main experiments were performed on a custom data split to ensure strict training-test separation (due to potential data overlap with Boltz-1's pretraining set), we include this additional benchmark to provide a broader evaluation. To ensure a fair comparison, all baseline models were retrained using our training data, so that any potential pretraining-related data leakage is equally accounted for across methods.

As shown in Table S8-S9, our model achieves consistently superior performance over all baseline methods on the RAbD benchmark across multiple evaluation metrics. These results further demonstrate the robustness and general applicability of our approach beyond the initially test set.

## E.8 Statistical Results

To assess the statistical reliability of our results, we report the standard error of the mean (SEM) for all main metrics in Tables S10-S12. The reported SEM values are consistently small across metrics, and the observed performance improvements of our method exceed the SEM margins by a clear margin, supporting the statistical significance and robustness of the reported gains.

## E.9 General Protein-Protein Structure Prediction

We evaluated MFDesign-C on 128 pure protein complexes from the Boltz-1 test set to assess its generalization. The model achieves comparable performance (as shown in Table S13) to Boltz-1 in terms of TM-score, RMSD, and DockQ.

Table S13: Structure prediction performance on protein complexes from the Boltz-1 test set.

| Method | TM-score ↑ | RMSD / Å ↓ | DockQ>0.23 / % ↑ |
|---|---|---|---|
| Boltz-1 | 77.43 | 9.55 | 35.16 |
| MFDesign | 75.24 | 10.79 | 34.38 |

Table S10: Results on typical antibodies with simultaneously designed CDRs, reported as mean±SEM. **Best** and second-best results are highlighted.

| CDR | Method | AAR ↑ | RMSD ↓ | CDR | Method | AAR ↑ | RMSD ↓ |
|-----|--------|-------|--------|-----|--------|-------|--------|
| H1 | DiffAb | 60.92±1.31% | 1.52±0.08Å | L1 | DiffAb | 56.72±1.16% | 1.43±0.11Å |
|    | dyMEAN | 70.31±1.40% | 1.65±0.17Å |    | dyMEAN | 73.06±1.81% | 1.58±0.14Å |
|    | MFDesign-C | 74.89±0.88% | **1.48±0.08Å** |    | MFDesign-C | **82.98±1.10%** | **1.41±0.06Å** |
|    | MFDesign-D | **74.95±0.81%** | 1.61±0.07Å |    | MFDesign-D | **82.98±1.15%** | 1.65±0.06Å |
| H2 | DiffAb | 33.53±1.16% | 1.44±0.05Å | L2 | DiffAb | 55.08±1.19% | 1.21±0.09Å |
|    | dyMEAN | 66.38±1.70% | 1.47±0.16Å |    | dyMEAN | 76.04±2.01% | 1.23±0.14Å |
|    | MFDesign-C | 65.59±1.33% | **1.26±0.06Å** |    | MFDesign-C | **88.84±0.92%** | **1.00±0.05Å** |
|    | MFDesign-D | **67.54±1.26%** | 1.44±0.06Å |    | MFDesign-D | 87.81±0.96% | 1.15±0.05Å |
| H3 | DiffAb | 22.26±0.66% | 4.29±0.18Å | L3 | DiffAb | 43.03±1.04% | 1.80±0.27Å |
|    | dyMEAN | 34.69±1.02% | 6.15±0.23Å |    | dyMEAN | 52.69±1.73% | 1.59±0.12Å |
|    | MFDesign-C | 63.13±1.16% | **3.54±0.14Å** |    | MFDesign-C | 78.93±1.27% | **1.55±0.05Å** |
|    | MFDesign-D | **65.04±1.07%** | 3.71±0.14Å |    | MFDesign-D | **80.15±1.24%** | 1.69±0.05Å |

Table S11: Loop-region performance on typical antibodies, shown as mean±SEM. Standard errors support the robustness of the reported improvements.

| Method | Loop-AAR ↑ | Loop-RMSD ↓ | IMP ↑ |
|--------|-----------|-------------|-------|
| DiffAb | 15.86±0.58% | 5.03±0.20Å | 53.35±3.93% |
| dyMEAN | 20.80±1.21% | 7.84±0.26Å | 5.60±1.81% |
| MFDesign-C | 61.71±1.44% | **4.13±0.16Å** | **65.37±3.75%** |
| MFDesign-D | **63.38±1.34%** | 4.28±0.17Å | 59.16±3.87% |

Table S12: Nanobody results with simultaneously designed CDRs, reported as mean±SEM. dyMEAN is excluded as it does not support nanobody inputs. **Best** and second-best results are highlighted.

| Method | CDR-H1 | | CDR-H2 | | CDR-H3 | | Loop-AAR / % | Loop-RMSD / Å | IMP / % |
|--------|--------|--|--------|--|--------|--|-------------|---------------|---------|
|        | AAR / % | RMSD / Å | AAR / % | RMSD / Å | AAR / % | RMSD / Å | | | |
| DiffAb | 45.17±2.24 | 2.52±0.17 | 31.11±2.04 | **1.54±0.12** | 15.39±0.98 | 4.71±0.25 | 10.51±0.60 | 5.50±0.29 | 30.93±7.05 |
| MFDesign-C | 65.71±2.59 | **2.48±0.16** | 55.60±2.93 | 1.62±0.13 | 61.40±1.86 | **3.71±0.21** | **61.01±2.20** | **4.26±0.24** | **43.14±7.55** |
| MFDesign-D | **67.34±2.34** | 2.62±0.16 | **58.93±2.35** | 1.78±0.14 | **63.51±1.72** | 3.84±0.21 | 60.80±2.15 | 4.41±0.24 | 38.26±7.41 |

## E.10 Inference Speed

We report the average inference time of our model by evaluating 10 randomly selected samples on a single NVIDIA H100 GPU. The -1 and -5 suffixes denote whether the model generates 1 or 5 predictions per sample, respectively. As shown in Table S14, the inference time of our method is comparable to that of Boltz-1, with minimal differences observed. Notably, generating multiple predictions per sample benefits from parallel decoding, leading to a reduced average time per prediction.

Compared to other antibody co-design baselines, our model incurs higher computational cost due to its larger number of parameters and more complex architecture. However, this overhead is accompanied by improved performance, reflecting a practical trade-off between computational efficiency and generation quality.

Table S14: Average inference time per sample for different methods.

| Method | Time / s ↓ |
|--------|-----------|
| DiffAb | 4.0 |
| dyMEAN | 1.1 |
| Boltz1-1 | 84.7 |
| Boltz1-5 | 32.8 |
| MFDesign-1 | 85.4 |
| MFDesign-5 | 33.0 |

## E.11 Visualization of More Examples

In this section, we provide more visualizations of our results. In Fig. S2, we show the outcomes from simultaneously designing all CDRs of antibodies conditioned on co-crystal structures, as discussed in Sec. 4.1, with a focus on our designed CDR-H3. In Fig. S3, we present additional cases from Sec. 4.2, where sequences with unknown amino acid types in the CDRs of nanobodies are used as inputs, and our model is able to predict the structure while also predicting the sequences of the nanobody CDRs. Additionally, in Fig. S4, we illustrate some predicted complex structures by our model when given the complete antibody-antigen complex sequence as input. These structures here are all predicted using the model with a discrete implementation of sequence diffusion.

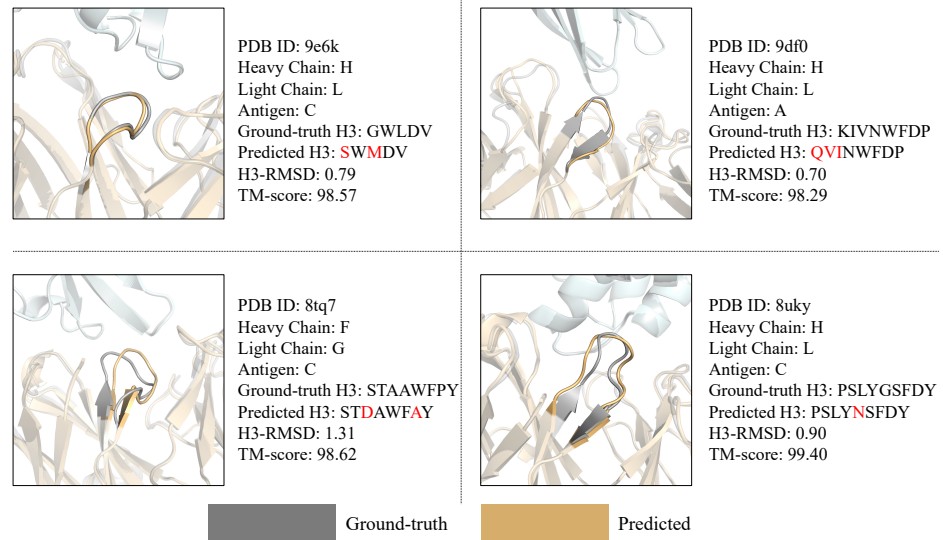

Figure S2: Examples of generated typical antibodies with simultaneously designed CDRs.

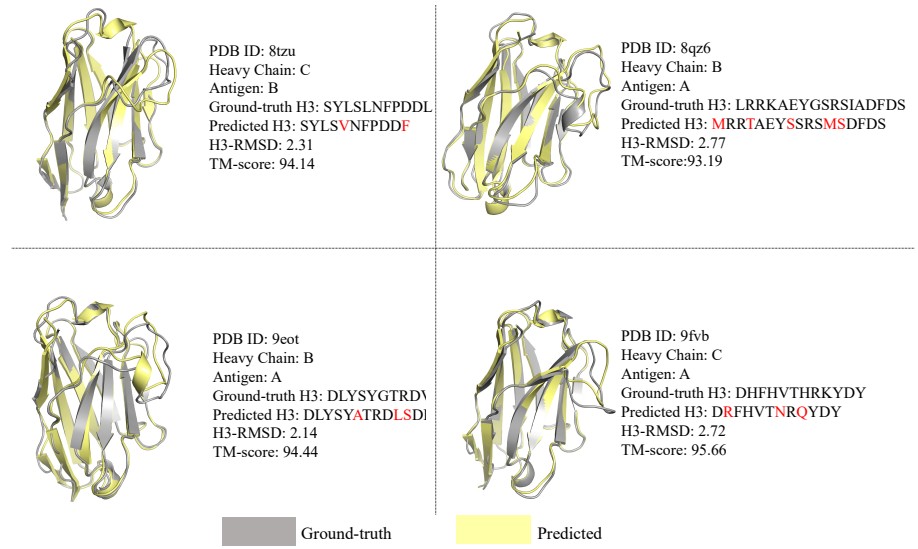

Figure S3: Examples of predicted nanobody structures using sequences with unknown CDRs as input.

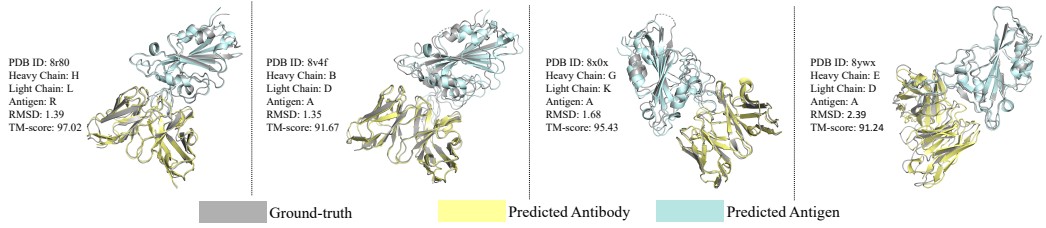

Figure S4: Examples of predicted antibody-antigen complex structure using complete sequences.

# F Correspondence of Modules Referenced in This Paper to AlphaFold 3 Algorithms

For direct technical cross-referencing, This section explicitly maps the AlphaFold3's modules referenced in this paper to their original algorithm numbers in the AlphaFold3's supplementary materials in Table S15.

Table S15: Mapping of referenced modules in this paper to AlphaFold3's algorithms.

| Module | AF3 Algorithm |
| --- | --- |
| **InputEmbedder** | Algorithm 2 |
| **TemplateModule** | Algorithm 16 |
| **MSAModule** | Algorithm 8 |
| **PairFormerStack** | Algorithm 17 |
| **DiffusionConditioning** | Algorithm 21 |
| **AtomAttentionEncoder** | Algorithm 5 |
| **DiffusionTransformer** | Algorithm 23 |
| **AtomAttentionDecoder** | Algorithm 6 |
| **DiffusionModule** | Algorithm 20 |
| **CentreRandomAugmentation** | Algorithm 19 |
| **RigidAlign** | Algorithm 28 |

# G Notations

We summarize the notations used in this paper in Table S16 to facilitate reading.

Table S16: Notations.

| Notation | Description |
| --- | --- |
| $<X>$ | a token which means the corresponding residue's type is unknown and requires to be speficied |
| $\mathbf{k}_i$ | a token, which is an one-hot encoding of its type |
| $\{\mathbf{k}_i\}$ | a token set |
| $\mathbf{x}_l$ | three-dimensional coordinates of a atom |
| $\{\mathbf{x}_l\}$ | a structure characterized by a set of atoms' coordinates |
| $\{\mathbf{f}^*\}$ | raw inputs |
| $\{\mathbf{s}_i^{\text{inputs}}\}$ | basic encodings for the raw inputs, generated by **InputEmbedder** in the conditioning trunk |
| $\{\mathbf{s}_i^{\text{trunk}}\}$ | token-level single representations, the output of the conditioning trunk |
| $\{\mathbf{z}_{ij}^{\text{trunk}}\}$ | token-level pair representations, the output of the conditioning trunk |
| $\{\mathbf{s}_i\}$ | token-level single representations generated by **DiffusionConditioning**, which embed information from the noise level, $\{\mathbf{s}_i^{\text{inputs}}\}$ and $\{\mathbf{s}_i^{\text{trunk}}\}$ |
| $\{\mathbf{z}_{ij}\}$ | token-level pair representations generated by **DiffusionConditioning**, which embed information from relative position encoding and $\{\mathbf{z}_{ij}^{\text{trunk}}\}$ |
| $\{\mathbf{x}_l^{\text{noisy}}\}$ | noisy structure |
| $\{\mathbf{r}_l^{\text{noisy}}\}$ | scaled $\{\mathbf{x}_l^{\text{noisy}}\}$ with approximately unit variance |
| $\{\mathbf{q}_l\}$ | atom-level single representations generated by **AtomAttentionEncoder**, which encode basic atomic features and incorporate information from $\{\mathbf{r}_l^{\text{noisy}}\}$ |
| $\{\mathbf{c}_l\}$ | atom-level single representations generated by **AtomAttentionEncoder**, which encode basic atomic features and incorporate information from $\{\mathbf{s}_i^{\text{trunk}}\}$ |
| $\{\mathbf{p}_{lm}\}$ | atom-level pair representations generated by **AtomAttentionEncoder**, which encode basic atomic features and incorporate information from $\{\mathbf{z}_{ij}\}$ |
| $\{\mathbf{a}_i\}$ | token-level single representations generated by **AtomAttentionEncoder** and updated by **DiffusionTransformer** |
| $\{\mathbf{x}_l^{\text{denoised}}\}$ | denoised structure |
| $\sigma_{\text{data}}$ | hyper-parameters for structure sampling |
| $\sigma_{\min}$ | hyper-parameters for structure sampling |
| $\sigma_{\max}$ | hyper-parameters for structure sampling |
| $\rho$ | hyper-parameters for structure sampling |
| $\gamma_0$ | hyper-parameters for structure sampling |
| $\gamma_{\min}$ | hyper-parameters for structure sampling |
| $\lambda$ | hyper-parameters for structure sampling |
| $\eta$ | hyper-parameters for structure sampling |
| $t$ | a sampled timestep |
| $T$ | the total number of timesteps for our sequence and structure diffusion training |
| $m_t$ | the mask rate for sequence diffusion at timstep $t$ |
| $\{\mathbf{k}_i^{\text{noisy}}\}$ | noisy sequence |
| $\{\mathbf{k}_i^{\text{denoised}}\}$ | denoised sequence |
| $\{\mathbf{x}_l^{\text{GT}}\}$ | given ground-truth co-crystal structure |

