# OpenReview forum: "Repurposing AlphaFold3-like Protein Folding Models for Antibody Sequence and Structure Co-design"
_NeurIPS.cc/2025/Conference — NeurIPS 2025 poster_

### Official Review · Reviewer_RhHw · 2025-07-01

**Clarity:** 3
**Significance:** 2
**Originality:** 1
**Rating:** 4
**Confidence:** 5

**Summary:**

This paper presents a method for antibody co-design by repurposing a large, pre-trained protein folding model (Boltz-1). The authors' approach, named MFDesign, augments the native structure diffusion module of the folding model with a new sequence diffusion head, enabling simultaneous co-design of antibody CDR sequence and structure. After fine-tuning this modified architecture on a relatively small dataset of antibody-antigen complexes, the model achieves dramatic performance improvements over state-of-the-art baselines while retaining its original structure prediction capabilities.

**Questions:**

* Discrete vs. Continuous Diffusion Trade-off: The results show a clear and interesting trade-off: the continuous diffusion model excels at structure (RMSD), while the discrete version excels at sequence (AAR). This is a valuable insight. Could you elaborate on this? Further, have you considered a hybrid approach that might leverage the strengths of both to achieve superior performance on all metrics?

* Impact of MSA Filtering: Your rigorous filtering of MSA data to prevent leakage is a major strength. To better quantify the importance of this step, could you report the model's performance on the test set without this filtering? This would highlight the extent of the data leakage problem in this domain and underscore the validity of your evaluation protocol. A strong response here would further increase my confidence in the work's contribution.

* What is the purpose of using AlphaFold3 if only modeling backbone?

**Ethical Concerns:**

["NO or VERY MINOR ethics concerns only"]

**Final Justification:**

I have raised my score and now recommend acceptance for this paper. The authors provided an exceptionally thorough and insightful rebuttal that successfully addressed my main technical concerns.

**Technical Soundness:** My primary concerns regarding the methodology have been fully resolved. The authors' new experiments (e.g., on MSA filtering) and their deep analysis (e.g., on the discrete vs. continuous diffusion trade-off) were highly convincing and demonstrated the technical solidity of this work.

**Positioning of Contribution:** A minor point remains regarding the precise positioning of the paper's core contribution. While I agree with the "foundation model" paradigm, this concept was already well-established in the AF2 era (e.g., by RFdiffusion). It would be beneficial for the paper to further articulate the specific, fundamental advantages of using an AF3-like model as the foundation (beyond general accuracy improvements), especially given the acknowledged limitation of not achieving all-atom generation.

**Limitations:**

Yes. The authors provide a clear and honest "Limitations and Future Work" section in the appendix. They appropriately identify the need for a separate side-chain packing step and the high computational cost of full-parameter fine-tuning as key limitations. They propose clear and reasonable directions for future research to address these points.

**Paper Formatting Concerns:**

No Paper Formatting Concerns.

**Quality:**

2

**Strengths And Weaknesses:**

**Strengths:**

The paper is very clearly written, presenting a complex idea in an accessible way. The methodology is well-explained, building logically on the known architecture of AlphaFold3. The evaluation is comprehensive, covering both typical antibodies and challenging nanobodies, and it includes an important analysis showing that the model's original folding capability is preserved after fine-tuning.

**Weaknesses:**

* A  weakness is that the method generates backbone atoms only, requiring a separate step with PyRosetta for side-chain packing and relaxation. While this is common practice, a fully end-to-end all-atom generation would be ideal.

* The novelty of the approach is questionable. The benefits of using AlphaFold 3 are not apparent for the task of backbone generation alone, and the core concept of applying discrete diffusion to protein structure prediction has already been demonstrated by DiffAb.

---

> ### Author Rebuttal · Authors · 2025-07-31
>
> We appreciate your valuable feedback. Below are our detailed responses. We hope they address your concerns and welcome your favorable reconsideration.
>
> ---
> > **W1 & L: All-atom generation**
> - Thank you for your question. Regarding all-atom generation, we recognize this limitation (Appendix C.1) and consider it a future research direction.
> - We would like to clarify that models like AF3 can directly determine the atoms contained in the side chain because the amino acid type is explicitly known at input time. However, in our model's input, the amino acid types of the CDRs part are unknown, so it is non-trivial to directly design an all-atom result when design sequence and structure when co-designing the structure and sequence.
> - Furthermore, many works, including AbX [1], also generate only backbone atoms and use Rosetta for side-chain packing and relaxation.
>
> > **W2 & Q3: Novelty**
> - Thanks for the critical question regarding the novelty of our work. This allows us to clarify our core contribution, which lies not in an innovation in a single technical component, but in proposing and validating a new and, we believe, more efficient paradigm for protein design.
> - We would like to clarify that our purpose for using AF3-like models is not merely to leverage their ability to generate protein backbones. Our core motivation is to "repurpose" large-scale folding models, which have been pre-trained on vast amounts of general protein data, as "Foundation Models" to solve the problem of data scarcity in specific downstream tasks, such as antibody design.
> - This idea is analogous to the paradigm in the field of Large Language Models (LLMs), where foundation models like GPT are fine-tuned for various downstream tasks. We posit that AF3-like models are, in effect, the foundation models for the "AI for Science" domain, and our work serves as a successful application of this "pre-train and fine-tune" paradigm in the field of antibody design. This is fundamentally different in principle from previous methods which train models from scratch on limited antibody data. We believe that this approach of "repurposing" protein folding foundation models holds great potential for the broader field of protein design in the future.
>
> > **Q1: Discrete vs. Continuous Diffusion Trade-off**
> - Thank you very much for the question and attention to this interesting finding in our work. We completely agree that this is a valuable observation and are happy to share our understanding and speculation on it.
> - First, we would like to clarify a premise: the "discrete" and "continuous" methods we compare are applied only to the generation of the sequence. For structure generation, we consistently use the native, continuous diffusion-based EDM method from the AF3-like model. Based on this, our explanation for the trade-off phenomenon you observed is as follows:
>     - **Why is discrete diffusion superior for sequence (AAR)**? We believe this primarily stems from data modality matching. Amino acid sequences are fundamentally discrete data (a choice from 20 categories). Discrete diffusion models are specifically designed to handle such data, allowing them to more accurately capture the underlying distribution and thus achieve better results in sequence recovery accuracy (AAR).
>     - **Why does using continuous diffusion (for sequence) lead to superior structure (RMSD)?** We speculate this is due to training schedule alignment, not a direct benefit to the structure.
>       - When we use continuous diffusion for the sequence, we can perfectly reuse the training schedule from the base model's (Boltz-1) pre-training, where the noise level is sampled from a continuous distribution. This maximally preserves the original model's structure prediction capability, hence resulting in better RMSD performance.
>       - When we use discrete diffusion for the sequence, we must "adapt" the structure module's continuous training schedule to discrete time steps for synchronization. This means the sampling of the noise level σ degenerates from a continuous space to a discrete set. This "non-native" training approach has a slight mismatch with the base model's pre-training, which we speculate leads to a minor degradation in structure prediction capability, manifesting as a slightly higher RMSD.
> - Based on the analysis above, an ideal "hybrid method" is not a simple combination of the two models, but rather one that resolves the training schedule mismatch while maintaining the advantage of discrete diffusion for sequence modeling.
> - **A promising direction is to adopt a discrete diffusion model that can operate in continuous time/schedule.** This would allow sequence and structure generation to be perfectly synchronized within a unified, continuous training framework, with each component operating in its optimal modality. We acknowledge that this is an advanced and challenging research direction with few available references at present. We have included the exploration of such methods as an important part of our future work and will state this in the paper. Thank you again for your insightful question.
>
> > **Q2: Impact of MSA Filtering**
> - Thank you for your insightful suggestion. We have conducted the corresponding test according to your feedback, and the results have led us to a more nuanced understanding, which we will elaborate on below.
> - Our methodology employs a 0.2 similarity threshold to filter the MSA. We would like to humbly clarify our rationale for this protocol. Our primary goal was to establish an academically rigorous and conservative benchmark from the outset. This strict filtering serves two purposes: 1) to preemptively prevent any potential for data leakage to interfere with our results, and 2) to ensure a consistent data distribution is used for both training and testing, which is fundamental for a reliable evaluation. In fact, we believe that even without filtering, the impact and risk of data leakage are minimal for the following reasons:
>   - Our statistical analysis shows that sequences with high similarity to the ground truth are extremely rare. For instance, sequences with >0.6 similarity constitute only about 7% of the raw MSA, and those with >0.8 similarity are a mere 2%.
>   - The `MSAModule` in our architecture internally subsamples the vast MSA for each computation. This mechanism further diminishes the already low probability of the model actually utilizing these few highly similar sequences.
>   - This concern is primarily relevant to the controlled setting of academic benchmarking to ensure fair comparison. In practical, real-world antibody design applications, leveraging all available homologous information is standard practice.
> - Therefore, we hypothesized that **data leakage would not have a significant impact** and the model would have difficulty learning from such a minuscule amount of "leaked" information. Consequently, we predicted that using unfiltered sequences as input would likely not lead to a substantial improvement in results.
> - We must be transparent that this is a theoretical conclusion based on the model's mechanics, as experimentally verifying it across various thresholds would be prohibitively expensive due to high training costs. After establishing our perspective, we performed the test as you suggested. The AAR(%) results are as follows:
>
> |Total|H1|H2|H3|L1|L2|L3|
> |-|-|-|-|-|-|-|
> |32.31±0.58|7.52±0.63|21.09±1.35|43.65±1.10|40.96±1.01|30.13±1.18|31.33±1.14|
>
> - As shown, the overall performance is indeed far worse than reported in our paper. We believe the primary and most plausible reason for this decline is **distribution shift**. To illustrate, the total unfiltered MSA is, on average, nearly **twice** as large as the one filtered with our strict 0.2 threshold. This dramatic increase in sequence quantity does not merely add more data; it also alters the information density of the input. It fundamentally alters the statistical properties of the input distribution our model was trained on, consequently impacting the distribution of key latent variables within the model.
> - However, it is crucial to note that the overall AAR is still above our 0.2 threshold, and the performance on the key CDR-H3 region remains at a respectable 43.65%. This strongly suggests that our model learns from complex patterns and multiple data (e.g., antigen's information), rather than simply "cheating" via leakage.
> - This drop demonstrates that violating distribution consistency hurts performance. This is precisely where the effectiveness and importance of our filtering method lie: it is the mechanism that establishes and guarantees this crucial consistency. While the 0.2 threshold may not be theoretically optimal—as a comprehensive search is too costly—our results in our paper empirically prove that it creates a sufficiently effective distribution, while also preventing potential leakage at source.
> - Tackling this sensitivity to distribution shift is a valuable direction for future work. We plan to explore potential solutions, such as training the model with filtered MSAs using random threshold to improve its robustness. This requires substantial experimentation. Due to time constraints, we are committed to pursuing this in our future research.
> - On the other hand, we do not view this distribution shift as entirely negative. The fact that respectable performance is maintained on key metrics suggests that this shifted distribution contains rich, relevant information, which could potentially be harnessed to generate more diverse protein designs, although this too would require further experimental validation.
>
> > **Reference**:
> - [1] Zhu, Ren, et al. “Antibody Design Using a Score-based Diffusion Model Guided by Evolutionary, Physical and Geometric Constraints.” ICML (2024).
> ---
> We hope this response could answer your questions and address your concerns, looking forward to receive your further feedback soon.

---

> > ### Comment · Reviewer_RhHw · 2025-08-05
> >
> > Thank you to the authors for the extremely detailed and insightful rebuttal, which has resolved most of my concerns. I find this to be a technically solid piece of work; therefore, I am raising my score and will support its acceptance. On that basis, I would like to pose one discussion point to further clarify the paper's contribution: Given that works like RFdiffusion already validated the paradigm of "repurposing prediction models for design" in the AF2 era, what is the core incremental advantage of upgrading the foundation model from AF2 to AF3, under the current limitation of not achieving all-atom generation? This is not a criticism, but a point to help better differentiate the paper from existing work and position its unique value.

---

> > > ### Author Response · Authors · 2025-08-06
> > >
> > > Thank you for your thoughtful comments and encouraging support. We are pleased that our response addressed your concerns. Your support is very encouraging to us. We hope our additional clarifications below will better address your concerns.
> > >
> > > > **The core incremental advantage of upgrading the foundation model to AF3**
> > > - We completely agree that the paradigm of "repurposing prediction models for design" has already been validated in the AF2 era by pioneering works like RFdiffusion. We would like to humbly clarify that, our work builds upon this foundation and offers several critical, incremental advantages, even under the current limitation of not achieving all-atom generation.
> > >
> > >   - First, compared to structure prediction models of the AF2 era (like AF2 and RoseTTAFold), AF3 represents a fundamental architectural shift. AF3-like models have a natively integrated diffusion module for structure prediction at their core, which was absent in AF2-era models. While design models like RFdiffusion are adapted from prediction models, their core generative module—the diffusion module—was built and trained additionally. Therefore, our work brings a more coherent architecture; the deep representations learned by the model's trunk were already optimized during pre-training to drive the diffusion process, ensuring a high degree of matching between features and the generative task. In contrast, the RFdiffusion paradigm relies on a more decoupled system, where the diffusion model is an additional component built upon a separate trunk (RoseTTAFold), which by design lacks the same level of deep integration. Furthermore, with such a native diffusion module, our method is also more concise and efficient from an engineering perspective, offering greater flexibility and scalability, while also not requiring the diffusion module to be retrained from scratch.
> > >   - Second, our model represents a significant paradigm shift. Works like RFdiffusion focus only on structure generation, requiring additional models like ProteinMPNN for subsequent sequence design in a multi-stage, step-by-step process. Our method, however, achieves true co-design by integrating sequence and structure diffusion within the same diffusion model. This better captures the coupling relationship between CDR sequence and structure, which is crucial for antibody design. In our response to Reviewer unJA, we also elaborated on the importance and necessity of this paradigm shift from multi-stage to co-design, which you may refer to.
> > >   - Finally, compared to previous structure prediction models, AF3 possesses superior foundational knowledge. It not only covers a broader range of PDB data but also includes interactions between various molecules like ligands, giving it a more extensive and fundamental understanding of biomolecular interactions. This general interaction knowledge can be effectively transferred to our antibody design task. Additionally, by fine-tuning on such a powerful foundation model, we can more effectively overcome the problem of data scarcity for antibody-antigen complex structures.
> > >
> > > - Lastly, regarding the limitation of all-atom generation that you mentioned, this is precisely a point that corroborates the architectural and paradigmatic advantages we described above.
> > >   - In terms of paradigm, we adopted the more powerful co-design setting, which is more difficult to implement for all-atom generation. This is because a multi-stage design like RFdiffusion's can generate the final side-chain structures after the sequence has been determined. In our model, however, the final amino acid types are difficult to determine before the diffusion process ends, making all-atom design a much more complex and challenging task.
> > >   - In terms of architecture, we fine-tune the native diffusion module rather than training an additional, extended diffusion module from scratch like RFdiffusion. This requires us to respect the training paradigm of the native diffusion module. The original AF3-like model can only predict the backbone atoms when handling unknown amino acids `<X>`. If we were to forcibly introduce all-atom generation, it would be a significant deviation from this pre-trained behavior and could potentially damage the valuable and rich foundational knowledge we hope to utilize.
> > > - Therefore, our focus on backbone generation is a strategic choice to prioritize the benefits of our co-design approach. We are fully aware of the limitations of a backbone-only approach and plan to investigate end-to-end all-atom generation in our future work. In conclusion, our work's upgrade to an AF3-like foundation is not merely an incremental replacement of the model but enables a more elegant, powerful, and biologically relevant co-design paradigm. We are very grateful for the opportunity to clarify this and will highlight these distinctions more clearly in the final version of our paper to position the unique value of our work.

---

### Official Review · Reviewer_jMg6 · 2025-07-03

**Clarity:** 3
**Significance:** 3
**Originality:** 3
**Rating:** 5
**Confidence:** 3

**Summary:**

This work proposes MFDesign, a method for modifying a AF3-like model for antibody sequence and structure co-design. MFDesign is equipped with an additional sequence diffusion head so that structure and sequence of antibody CDRs can be co-generated. For this, the authors re-use AF3's structural RDM diffusion, add a discrete (MFDesign-D) or continuous (MFDesign-C) sequence diffusion path, align both paths, and fine tune on ~6k antibody-antigen complexes from SAbDab.
The authors show that MFDesign outperforms compared-to methods on antibody co-design tasks on SabDab. While no benchmarks on non-antibody complexes were conducted, MFDesign maintains the original folding accuracy of AF3 for antibody complexes.

**Questions:**

In addition to the questions/points in the Strengths And Weaknesses section:
19. Please provide estimates for statistical confidence for the reported improvements.
20. It would be beneficial for practical application to include a rough estimate of the computational effort with a comparison to the compared-to methods. Please include that if possible.

**Ethical Concerns:**

["NO or VERY MINOR ethics concerns only"]

**Final Justification:**

## Resolved
1. Added uncertainty estimates (SEMs), a second benchmark on RAbD with retrained baselines, per-sample runtime on H100, and clarified/quantified the >2k-residue limit (rare; one test case). These directly address my main concerns about robustness, fairness, and practicality.
2. Partially resolved: For generalization beyond antibodies the authors report comparable folding on 128 non-antibody complexes. Full co-design beyond antibodies remains future work but this is helpful, so I view it as "partially resolved".

## Unresolved
3. No wet-lab validation.
4. Current sequence diffusion supports only standard residues.
5. Paired statistical tests (e.g., bootstrap CIs over the same complexes) would be helpful despite small SEMs.

## Weighting/Conclusion
The consistent gains on key antibody co-design metrics and maintained folding accuracy, plus improved transparency (data curation, curriculum, compute), in my opinion, outweigh the unresolved limitations (which are also common in the area).
This leads me to recommend "Accept", assuming the rebuttal changes are integrated in the final version.

**Limitations:**

The authors do address some limitations (MSA leakage) but do not discuss that in-silico metrics may not translate to actual binding affinity and that MFDesign inherits AF3's limitations regarding novel/non-standard residues.

**Paper Formatting Concerns:**

No concerns

**Quality:**

3

**Strengths And Weaknesses:**

# Quality
## Strengths

1. The paper filters and splits the data well, with meaningful considerations for train-test leakage. In my opinion, the time-aware split of SAdDab is a very good approach and the other filters, e.g. deduplicating using Levenshtein distance and MSA scan, make sense. I also appreciated the clear description of the filtering steps in the appendix e.g. (Algorithm S1).
2. The method is transparent and the 4 stages of the curriculum are outlined well with supporting evidence for their usefullness via abalation study.
3. The paper reports multiple metrics: AAR, RMSD, Loop metrics, IMP, and Rosetta.


## Weaknesses

4. Missing statistical uncertainty estimates make it difficult to put the results into perspective. E.g. if the large AAR gain is truly significant.
5. While the data curation is well done, the paper seems to rely on a single benchmark - which is restricted to the authors' SABDab split. I can understand why RAbD60 was discarded but this still opens the door for unaccounted affects that may distort the results. In my opinion, it would benefit the paper to have another benchmark.
6. It would have been great to have proper comparison to AbDiffuser, AbX, IgGM, and RFdiffusion-Antibody. I appreciate the authors reasoning why they were omitted (and missing code is a hard-stop for complex models) but I will list it here for completeness.
7. There seem to be inherent resource limitations, which led to filtering out antibody-antigen complexes that exceed 2k residues. How sever is this? It's only mentioned in the appendix (I honestly might have by chance overread it), is it justified to only mention it there?

# Clarity
## Strengths
8. The list of contributions is clear and easy to read.
9. The Algorithm 1 part clarifies the implementation.
10. The authors seem to be transparent and provide information on training objective, continuous and discrete diffusion, as well as hyperparameters in the appendix.
11. Similar for the data pipeline - documented step-by-step in the appendix.


# Significance
## Strengths
12. I think this work is significant because of the potential practical use in therapeutic design. Co-design of CDR sequence/structure, conditioned on antigen pose, seems relevant for industry (if results hold in real-world applications).
13. The reported improvements over the compared-to methods seem substantial.
14. The preserved folding accuracy is interesting and might serve as a nice evidenced example for the benefits of transfer-learning.

## Weaknesses
15. Unfortunately, the restriction to in-silico experiments only does take away from the significance of the results (which is not uncommon in DL papers for budget reasons).
16. As mentioned above, the reliance on a single dataset is in my opinion reducing the strength of the results.
17. Given the maintained folding accuracy, it would be interesting to see the performance of MFDesign on general protein–protein-interface data.


# Originality
## Strengths
18. To my knowledge, the modification of AF3-style diffusion with discrete sequence diffusion for antibodies is novel.

## Possibly relevant missing literature references
I'm not affiliated with any of these missing references but they seem relevant to the paper:

Ingraham, John B., et al. "Illuminating protein space with a programmable generative model." Nature 623.7989 (2023): 1070-1078.

Lisanza, Sidney Lyayuga, et al. "Multistate and functional protein design using RoseTTAFold sequence space diffusion." Nature biotechnology (2024): 1-11.

---

> ### Author Rebuttal · Authors · 2025-07-30
>
> Thank you for dedicating your time and providing valuable feedback. In the following, we have carefully crafted detailed responses to address your comments.
>
> ---
> > **W4 & Q19: Statistical Confidence**
>
> - Thank you for your suggestion. To ensure the credibility of our results, we have added the standard error of the mean (SEM) to our main results as you suggested. Here are the updated results for Tables 1-3:
>
> - Typical antibody results:
>
> | Method| H1 AAR(%)/RMSD(Å) | H2 AAR(%)/RMSD(Å) | H3 AAR(%)/RMSD(Å) | L1 AAR(%)/RMSD(Å) | L2 AAR(%)/RMSD(Å) | L3 AAR(%)/RMSD(Å) | Loop-AAR(%)/RMSD(Å) | IMP(%) |
> | - | -|- | - | - | - | - | - | - |
> | DiffAb | 60.92±1.31/1.52±0.08 | 33.53±1.16/1.44±0.05 | 22.26±0.66/4.29±0.18 | 56.72±1.16/1.43±0.11 | 55.08±1.19/1.21±0.09 | 43.03±1.04/1.80±0.27 | 15.86±0.58/5.03±0.20 | 53.35±3.93 |
> | dyMEAN | 70.31±1.40/1.65±0.17 | 66.38±1.70/1.47±0.16 | 34.69±1.02/6.15±0.23 | 73.06±1.81/1.58±0.14 | 76.04±2.01/1.23±0.14 | 52.69±1.73/1.59±0.12 | 20.80±1.21/7.84±0.26 | 5.60±1.81 |
> | MFDesign-C | 74.89±0.88/**1.48±0.08** | 65.59±1.33/**1.26±0.06** | 63.13±1.16/**3.54±0.14** | **82.98±1.10**/**1.41±0.06** | **88.84±0.92**/**1.00±0.05** | 78.93±1.27/**1.55±0.05** | 61.71±1.44/**4.13±0.16** | **65.37±3.75** |
> | MFDesign-D | **74.95±0.81**/1.61±0.07 | **67.54±1.26**/1.44±0.06 | **65.04±1.07**/3.71±0.14 | **82.98±1.15**/1.65±0.06 | 87.81±0.96/1.15±0.05 | **80.15±1.24**/1.69±0.05 | **63.38±1.34**/4.28±0.17 | 59.16±3.87 |
>
> - Nanobody results:
>
> | Method     |      H1 AAR(%)/RMSD(Å)       |    H2 AAR(%)/RMSD(Å)     |      H3 AAR(%)/RMSD(Å)       |  Loop-AAR(%)/RMSD(Å)  |     IMP(%)     |
> | - | - | - | - | - | - |
> | DiffAb     |     45.17±2.24/2.52±0.17     | 31.11±2.04/**1.54±0.12** |     15.39±0.98/4.71±0.25     |   10.51±0.60/5.50±0.29   |   30.93±7.05   |
> | MFDesign-C | 65.71±2.59/**2.48±0.16** | 55.60±2.93/1.62±0.13 | 61.40±1.86/**3.71±0.21** | **61.01±2.20**/**4.26±0.24** | **43.14±7.55** |
> | MFDesign-D |     **67.34±2.34**/2.62±0.16     |   **58.93±2.35**/1.78±0.14   |     **63.51±1.72**/3.84±0.21     |   60.80±2.15/4.41±0.24   |   38.26±7.41   |
>
> - The standard errors (SEM) are small across all metrics, and our performance gains significantly exceed the SEM, confirming the statistical significance of our improvements.
>
> > **W5 & 16: Another benchmark**
>
> - We thank you very much for your valuable feedback regarding benchmark diversity. We completely agree that validation on multiple benchmarks greatly enhances the reliability and generalizability of research findings.
>
> - Our initial decision to create our own data split was driven by a strict consideration for experimental fairness. As you understood, since our method is based on a model (Boltz-1) with a specific pre-training data cut-off date, data from many public benchmarks (like RAbD) have already been "seen" by it. A direct test would lead to an unfair evaluation due to data leakage.
>
> - Nevertheless, we recognize the limitations of a single benchmark. Therefore, in response to your suggestion, we have supplemented our work with a full experiment on the RAbD benchmark. For fairness, all baselines were retrained on our training set, ensuring a comparable level of potential data leakage for all models.
>
> - The new experimental results on the RAbD benchmark are as follows:
>
> | Method | H1 AAR(%)/RMSD(Å) | H2 AAR(%)/RMSD(Å) | H3 AAR(%)/RMSD(Å) | L1 AAR(%)/RMSD(Å) | L2 AAR(%)/RMSD(Å) | L3 AAR(%)/RMSD(Å) | Loop-AAR(%)/RMSD(Å) | IMP(%) |
> |  - |  - |  - |  - |  - |  - |  - |  - |  - |
> | DiffAb | 67.03±2.25/1.10±0.07 | 35.16±1.36/1.13±0.06 | 26.66±1.30/2.88±0.17 | 57.95±1.75/0.96±0.04 | 55.84±1.80/0.92±0.04 | 45.15±1.35/1.21±0.07 | 20.21±1.32 / 3.44±0.19 | 22.29±5.42 |
> | dyMEAN | 79.15±2.32/**1.03±0.08** | **73.87±2.82**/0.97±0.05 | 38.46±2.01/4.44±0.20 | 77.00±1.81/0.97±0.05 | 88.30±3.28/0.79±0.04 | 57.88±2.70/1.20±0.09 | 24.28±2.68 / 6.00±0.24 | 0.00±0.00 |
> | MFDesign-C | 79.06±1.16/1.08±0.06 | 70.93±1.91/**0.89±0.04** | 77.87±1.93/**2.26±0.13** | **86.80±1.38**/**0.87±0.04** | **92.85±1.20**/**0.68±0.02** | **86.17±1.47**/**1.12±0.07** | 73.76±2.82 / **2.69±0.15** | **48.14±0.07** |
> | MFDesign-D | **79.55±1.13**/1.15±0.07 | 72.31±2.01/1.04±0.04 | **78.61±1.69**/2.35±0.12 | 86.31±1.52/1.12±0.05 | 90.46±1.39/0.80±0.02 | 85.26±1.73/1.24±0.07 | **75.27±2.43** / 2.75±0.14 | 41.95±6.42 |
>
> - As shown in the table above, our model remains comprehensively and significantly superior to all baseline methods. This added result further demonstrates the robustness and superiority of our method. We will include this experiment and its discussion in the final version of the paper. Thank you again for your suggestion, which has made our work more complete.
>
> > **W6: Another baseline**
>
> - Thank you for your recognition. As we pointed out in the paper, among the mentioned papers:
>
>     * AbDiffuser has no open-source code.
>     * IgGM and AbX only released inference code and model weights, without training code.
>     * RFAntibody adopts a two-stage method: first generating the structure using RFDiffusion, and then generating the sequence using ProteinMPNN. This is fundamentally different from our co-design settings.
>
> - Therefore, we did not include them in the main results in the main text. However, to provide results that are as rich as possible, we still provided a comparison of the results with IgGM and RFAntibody in Tables S2, S3, and S4 in appendix, which you can refer to. Our model still has an advantage in the results.
>
> > **W7: Filter out complexes that exceed 2k residues**
>
> - We thank you for your meticulous review and valuable suggestion. Regarding the severity of this limitation, we believe its impact is minor. In our final test set, only a single antibody-antigen complex (PDB ID: 8sak) was filtered out due to this limit. We consider its impact negligible for the following key reasons:
>
>     * Data Rarity: Antibody-antigen complexes larger than 2,000 residues are very rare in the entire PDB database. In practical applications, one could also significantly reduce the input length by using only the antigen's epitope region.
>
>     * Hardware Limitation: The limit stems from GPU memory constraints, not an intrinsic flaw in our algorithm. On hardware with larger GPU memory, our model could process these very long complexes without any modification.
>
>     * Existing Solutions: This limitation is inherited from our base model Boltz-1. It is worth noting that subsequent updates to Boltz have effectively mitigated this issue by introducing techniques like chunking. Migrating our method to these updated versions would likely support longer sequences even on existing hardware.
>
> - We agree and will move this disclosure to the main text in the final version for better visibility.
>
> > **W15 & L: Limitation of in-silico experiments and novel/non-standard residues**
>
> - Thank you for your suggestion. Due to cost issues, we are indeed unable to conduct wet-lab experiments to verify the actual results and can only measure the results through computer simulation experiments and related metrics. We want to state frankly that this limitation is common in the field.
>
> - Regarding the limitation of AF3 on novel and non-standard residues, we sincerely accept it. Our current discrete diffusion only supports the 20 standard types of amino acids. However, to our knowledge, Boltz has added support for Modified Residues in subsequent updates, thus being able to predict the structure of non-standard residues. Therefore, this limitation can be solved by migrating our method to the latest Boltz model and solving the sequence generation problem for non-standard residues. We will list it in the "Future work" section of our paper.
>
> > **W17: Performance of general protein–protein-interface data.**
>
> - Thank you for your suggestion. As our work primarily focuses on the sequence-structure co-design of antibodies, it cannot support the design of general proteins. We would like to clarify that our method can indeed maintain a certain folding accuracy. To test the structure prediction capability of MF-Design on general proteins, we took a total of 128 pure protein complexes from the Boltz-1 test set for testing and compared them with Boltz-1 itself. We chose MFDesign-C because it has better structure prediction ability. The following are the comparison results:
>
> |Method|TM-score|RMSD(A)|DockQ>0.23(%)|
> |-|-|-|-|
> |Boltz-1| 77.43 | 9.55 | 35.16 |
> |MFDesign| 75.24 | 10.79 | 34.38 |
>
> - It can be found that on TM-Score, RMSD, and DockQ, our model still maintains folding accuracy comparable to Boltz-1.
>
> > **W20: Estimate of computational effort**
>
> - Thank you for your suggestion, we will add this result in the final version of paper. Below is an estimation of our running time, which is the average time to generate a single prediction by running 10 different sample inputs in single NVIDIA H100 GPU. The suffixes -1 and -5 represent predicting 1 result or 5 results per sample, respectively.
>
> |Method|DiffAb|dyMEAN|Boltz1-1 | Boltz1-5 | MFDesign-1 | MFDesign-5 |
> |-|-|-|-|-|-|-|
> |Time(s)|4.0|1.1|84.7|32.8 | 85.4 | 33.0 |
>
> - It can be found that the difference in running time between our model and Boltz-1 is almost negligible. Furthermore, when predicting multiple results per sample, it supports parallel prediction, which greatly speeds up the average running time for a single prediction result.
>
> - Compared to other baseline methods for antibody codesign, our method does not have an advantage in computational efficiency due to the larger number of model parameters and a more complex architecture. However, the advantages in parameter count and architecture also bring us better results, which we believe is a reasonable performance-efficiency tradeoff.
>
> > **Possibly relevant missing literature references**
>
> - Thank you for your additions. We will add these two related articles to our citations.
>
> ----
> We hope this response could answer your questions and address your concerns, looking forward to receive your further feedback soon.

---

> ### Comment · Reviewer_jMg6 · 2025-08-06
>
> I thank the authors for their very detailed and helpful rebuttal and the added experiments.
>
> In my opinion, this rebuttal strengthens the paper (added uncertainty estimates (SEMs), a second benchmark on RAbD with retrained baselines, computational cost via H100 timings, and clearer discussion of the >2k-residue limit). Assuming the above clarifications are incorporated into the final paper version, I change my initial "Borderline Accept" to an "Accept". The work is practically relevant, technically sound, and with the changes from the rebuttal also empirically stronger.

---

> ### Author Response · Authors · 2025-08-06
> **Thanks for your comments**
>
> We sincerely thank you for taking the time to review and thoughtfully engage with our rebuttal. We are pleased that the additional analyses address your concerns and further strengthen the work. We greatly appreciate your positive evaluation and your revised recommendation to accept. As noted, we will incorporate the above clarifications into the final version of the paper. Your constructive feedback and support are truly valuable and mean a great deal to us.

---

### Official Review · Reviewer_ppEN · 2025-07-03

**Clarity:** 2
**Significance:** 2
**Originality:** 3
**Rating:** 4
**Confidence:** 4

**Summary:**

The authors propose a method to repurpose the diffusion module part of AlphaFold3-like models for sequence-structure co-design of antibodies. In detail, the authors add a residue-type denosing part to the original diffusion module and using mask tokens as input to CDR regions to enable sequence-structure co-design. The authors provide a 4-stage training strategy to fine-tuning Boltz-1, an open-source version of AlphaFold3. The authors provide extensive experiments on antibody benchmarks, and a detailed ablation study alsat the same time.

**Questions:**

1. The demonstration of methodology looks quite heavy in the current manuscript. I hope the authors could help me make their contributions more clear.
2. One concern about fine-tuning AlphaFold3-like models on specific domains is the training strategy. The authors provide a very detailed description for their strategy, but I wonder how is this strategy determined? Will changing the strategy harm the performance of robustness of the model? Especially, will the structure prediction of antigens be harmed?
3. There are multiple choices for discrete token diffusion, and I wonder why did the authors choose the current version? It will be very interesting if there could be an ablation study about this.

**Ethical Concerns:**

["NO or VERY MINOR ethics concerns only"]

**Final Justification:**

The authors have addressed all my questions, making comprehensive ablation studies. Thus, my scores will remain unchanged as 4.

**Limitations:**

Yes.

**Quality:**

3

**Strengths And Weaknesses:**

The paper provides a very nice method to finetune AlphaFold3-like models. This approach is meaningful as it can maintain the good structure prediction ability of protein folding models while smoothly adding design ability. Such methods have strong potential in many protein-related domains other than antibodies. The experimental analysis is quite extensive within its scope, and addresses most concerns. While the paper is clearly written, the demonstration could be further improved. I think the methodology part is a bit heavy, and it is best to move the detailed model structure algorithm into the supplementary while keeping this paper's critical contribution clear.

---

> ### Author Rebuttal · Authors · 2025-07-30
>
> Thank you for acknowledging our work and the valuable feedbacks. In the following, we have carefully crafted detailed responses to address your comments.
>
> ---
> > **W & Q1: Heavy methodology writing and clearer contribution**
>
> - We sincerely appreciate your suggestion. We apologize for the issues with the writing. We promise to thoroughly revise the main text in future version, move non-essential parts to the appendix, and highlight our key contributions in the main paper.
>
> - We would like to humbly clarify that our contribution lies in transforming a well pre-trained AF3-like model (leveraging its learned general protein interaction knowledge) into an antibody sequence-structure co-design model, which effectively addresses the scarcity of antibody-antigen complex structure data while achieving state-of-the-art performance with original folding accuracy maintained, resulting a multi-purpose method. In addition, such paradigm may offer insights for other protein-related fields where similar specific data limitations exist beyond antibodies.
>
> > **Q2: Impact of fine-tuning strategies**
>
> - Thank you for this insightful question. Our training strategy is guided by the principle of Curriculum Learning, which is a well-established and effective methodology for training large-scale models. We were inspired by the multi-stage training paradigm of models like AlphaFold3, where the core idea is to present data from easy to hard, allowing the model to first learn local, simpler features before progressing to global, more complex structures.
>
> - Unlike a general-purpose model, we have tailored this curriculum specifically for the task of antibody co-design. Since our primary goal is to design the CDRs that bind to an antigen, we have designed the training to expand its focus outwards from a CDR-centric perspective. Specifically:
>
>     - Stages 1-3: The cropping strategy we have designed for the first three stages always includes the complete CDR regions of the antibody. We first let the model learn the antibody structure in isolation (Stage 1), and then gradually introduce the context of the antigen epitope and its surrounding areas (Stages 2 & 3), while progressively increasing the maximum number of input tokens. This design forces the model to continuously focus on the antibody-antigen interface, which is most critical to our task.
>
>     - Stage 4 & Robustness: We fully anticipated the risk you astutely pointed out: an excessive focus on the CDR-proximal regions might degrade the model's ability to predict the structure of antigen parts that are distant from the binding interface. To mitigate this, we have specifically designed a fourth training stage. In this stage, we have implemented a mixed-strategy cropper: there is a 50% probability of using the aforementioned CDR-centric cropping, and a 50% probability of performing completely random sub-structure sampling across the entire antibody-antigen complex. This ensures that while the model masters local design, it also revisits and maintains its pre-trained capability for global complex structure prediction.
>
> - The necessity of this strategy is empirically validated. As shown in our ablation study in Appendix G.2 (Table S1), without the fourth training stage, the model's prediction performance on the antigen structure worsens, as measured by both RMSD and TM-score. Therefore, our carefully designed four-stage strategy is crucial for the model's robustness.
>
> - Finally, given the substantial computational cost of fine-tuning models of this scale, an exhaustive search of all possible strategies was not feasible. However, we believe our proposed and empirically-validated curriculum learning strategy strikes an effective balance between task-specific adaptation and the preservation of the model's general capabilities.
>
> > **Q3: Ablation study with other discrete token diffusion method.**
>
> - Thanks for the valuable suggestion. First, we will explain why we chose D3PM-absorb as the version for sequence diffusion.
>
>     + D3PM [2], as a typical method for discrete data diffusion, has several different types, including uniform, absorb, gauss, etc. D3PM-absorb is the most suitable one for amino acid sequence design. Specifically, D3PM-absorb can be seen as a Masked Language Model, suitable for handling the generation of text sequence modality. Since amino acid sequences can be regarded as text modality information, they are applicable to D3PM-absorb.
>
>     + In addition, in the original D3PM paper, Table 1 also compared the performance of different types on text generation, and the effect of absorbing was significantly better than other types of discrete diffusion. This proves the effectiveness of D3PM-absorb in terms of performance.
>
>     + Furthermore, many previous protein sequence design works have adopted D3PM-absorb. For example, LaMBO-2 [1] from NeurIPS 2023 used D3PM-absorb for antibody sequence generation. Our specific implementation also referenced the implementation of LaMBO-2.
>
> - Next, we follow your valuable suggestions and choose another discrete token diffusion model D3PM-uniform to conduct further ablation studies. We have supplemented the results on the test set after retraining with D3PM-uniform for sequence diffusion, as shown below:
>
> - Typical antibody results:
>
> | Method              | H1 AAR(%)/RMSD(A) | H2 AAR(%)/RMSD(A) | H3 AAR(%)/RMSD(A) | L1 AAR(%)/RMSD(A) | L2 AAR(%)/RMSD(A) | L3 AAR(%)/RMSD(A) | Loop-AAR(%)/RMSD(A) | IMP(%)    |
> | - | - | - | - | - | - | - | - | - |
> | MFDesign-C          | 74.89/1.48            | 65.59/**1.26**        | 63.13/**3.54**        | **82.98**/1.41        | **88.84**/1.00        | 78.93/**1.55**        | 61.71/**4.13**      | **66.24** |
> | MFDesign-D(absorb)  | **74.95**/1.61        | **67.54**/1.44        | **65.04**/3.71        | **82.98**/1.65        | 87.81/1.15            | **80.15**/1.69        | **63.38**/4.28      | 59.66     |
> | MFDesign-D(uniform) | 71.82/**1.41**        | 55.74/1.29            | 50.15/3.70            | 77.09/**1.38**        | 82.47/**0.98**        | 73.06/**1.55**        | 48.99/4.30          | 64.72     |
>
> - Nanobody results:
>
> | Method              | H1 AAR(%)/RMSD(Å) | H2 AAR(%)/RMSD(Å) | H3 AAR(%)/RMSD(Å) | Loop-AAR(%)/RMSD(Å) | IMP(%)    |
> | - | - | - | - | - | - |
> | MFDesign-C          | 65.71/**2.48**    | 55.60/**1.62**    | 61.40/3.71        | **61.01**/4.26         | 42.67     |
> | MFDesign-D(absorb)  | **67.34**/2.62    | **58.93**/1.78    | **63.51**/3.84    | 60.80/4.41         | 38.26     |
> | MFDesign-D(uniform) | 60.08/2.51        | 46.29/1.65        | 45.45/**3.60**    | 43.88/**4.14**     | **43.84** |
>
>
> - The results show that D3PM-uniform is significantly weaker than D3PM-absorb in terms of AAR. On the other hand, its RMSD results do not show a clear advantage over continuous diffusion methods. This ablation study demonstrates the effectiveness of our chosen method.
>
> > **Reference**:
>
> - [1] Gruver, Nate, et al. "Protein design with guided discrete diffusion." NeurIPS (2023).
>
> - [2] Jacob Austin, Daniel D Johnson, et al. "Structured denoising diffusion models in discrete state-spaces." NeurIPS (2021).
>
> ---
> We hope this response could answer your questions and address your concerns, looking forward to receive your further feedback soon.

---

> > ### Comment · Reviewer_ppEN · 2025-08-09
> >
> > Thank you for your response. I really appreciate your ablation study which helps me understand this model better. I will remain as supportive as I was to this paper.

---

> > > ### Author Response · Authors · 2025-08-09
> > >
> > > Thank you for your kind feedback. We are truly glad to hear that our response and ablation study have adequately addressed your concerns. Your constructive comments have been invaluable in improving our paper, and we sincerely appreciate your continued support throughout this process.

---

### Official Review · Reviewer_unJA · 2025-07-04

**Clarity:** 3
**Significance:** 4
**Originality:** 4
**Rating:** 5
**Confidence:** 3

**Summary:**

This paper proposes adapting a pretrained Boltz1 for the codesign of antibody binders conditioned on antigens, making one of the first very performant breakthroughs (concurrent with chai2) in antibody co-design, The progress made by this work has the promise to be very impactful to the community in particularly if everything is open-sourced; thus I provide my positive score on the condition that everything is opened-sourced on camera-ready.

**Questions:**

1. My understanding is AF3 is an all atom model , meaning it predicts the side chains , once one has all the side chains the sequence/ aa type can be obtained, so why do we chose to go for co-design here ? is boltz-1 not all atom?

2. Whilst I agree that antibodies are very sparse in protein datasets such as the PDB, one could also try to pretrain a diffusion model on not only monomers but all available complexes in the PDB and then finetune on a smaller antibody dataset. In spirit this feels as data efficient as what the authors have done (I dont see the issue here other than the fact that maybe no such diffusion model is openly available at the moment).

3. As the work mentions Boltz1 and AF3 are conditional diffusion models typically conditioned on sequence (in this use case the sequence of the target antigen) they are then able to generate the structure, however these key and clear detail has been somewhat ommited from the way the authors discuss AF3 like models , they jump straight into talking about the trunk making it harder to parse for a wider ML auidience.

4. In theory AF3 like models have and can be used to generate antibodies conditioned on a antigen? from what I understand the model can take as input the sequence of the antigen ? happy to be corrected here , so why not include an AF3 like baseline with boltz1 ? why is the sequence codesign modification necessary ? Im a bit confused here

**Ethical Concerns:**

["NO or VERY MINOR ethics concerns only"]

**Final Justification:**

I'm happy with the reviewer's response, and I maintain my positive score.

**Limitations:**

I think in their appendix section on limitations this has been well addressed.

**Quality:**

3

**Strengths And Weaknesses:**

Strenghts

1. Ignoring extremely concurrent work (which this work should not be compared to) I think this method makes breakthrough strides in the problem of antibody design
2. The methodology /adaptations are mostly well explained, and the work requires a substantial piece of ML engineering / empirical work.

Weakness

1. This is a small writing preference, but why keep calling things AF3-like? It sounds a bit like PRy, Boltz1 is a well-known implementation with some of its own additional features which distinguish it from AF3 (e.g. Inference time steering, among other things).

2. Some details in the main should be a little clearer without needing to dive into the appendix (See questions)

---

> ### Author Rebuttal · Authors · 2025-07-30
>
> Thank you for acknowledging our work. Your rating has provided us with great encouragement, and your detailed feedback and suggestions have been immensely helpful. Below is our detailed response.
>
> ---
> > **W1: Why call it AF3-like**
>
> - We sincerely appreciate the reviewer for this valuable feedback on our terminology. We fully agree that Boltz-1 is an excellent model with its own distinguishing features, and our decision to use the term "AF3-like" is based on two primary considerations:
>
>   - As mentioned in our introduction, a series of models including Chai-1, Boltz-1, and Protenix have recently emerged. They share the core diffusion-based architecture pioneered by AlphaFold3. We use "AF3-like" to refer to this class of models that share a common technical foundation.
>   - Our core contribution — the integration of sequence-structure co-diffusion — is an architectural modification that applies to this general framework, not to any feature specific to Boltz-1. We chose Boltz-1 as it is an excellent open-source implementation with available training scripts. Therefore, the term "AF3-like" more accurately reflects the general applicability and potential impact of our work, rather than confining it to a single model.
>
> > **W2 & Q3: Clearer details in main article**
>
> - We sincerely appreciate the reviewer for this invaluable suggestion. Due to space limitations, we had to place some key details in the appendix. Following NeurIPS' standard practice, accepted papers are permitted one additional page - we would use this to incorporate the highlighted content. If accepted, we will add a clear introduction to more details, improving accessibility. Thank you again for your constructive feedback, which will significantly improve the quality of our paper.
>
> > **Q1: All-atom generation**
>
>
> - Thank you for this insightful question, which touches upon a core consideration of our methodology. We would like to clarify this with the following points:
>
>   - We have acknowledged this limitation in Appendix B.5 and consider it as one of our future research directions.
>   - Regarding the premise that sequence can be derived from predicted side chains, this inverts the actual operational mechanism of models like AF3. These models are conditioned on a known amino acid sequence to predict an all-atom structure, including the side chains. When the sequence is provided, the model knows which side-chain atoms to generate. However, in our co-design task, the CDR sequences are initially unknown and are represented by the special `<X>` token. In this context, the model cannot determine which side chains to build for these unknown residues, making direct end-to-end all-atom design a non-trivial problem.
>   - Our backbone-first approach is an established paradigm. Many existing works, including the recent AbX[1] (ICML 2024), also generate backbone atom coordinates first, followed by side-chain packing and relaxation using tools like PyRosetta.
>
> - We hope this addresses your concerns. In summary, while Boltz-1 is capable of all-atom prediction, our specific task setting justifies the rationality of our backbone-first strategy.
>
> > **Q2: Pretrain a diffusion model using all PDB and finetune on antibody dataset**
>
> - Your insight is exactly right – this is precisely what we do in our work! In fact, the paradigm used in this paper exactly follows a pretraining-finetuning approach as you proposed and we have already used a pretrained diffusion model. Boltz-1 is actually a pretrained AF3-like diffusion model, trained on the entire PDB, a large dataset containing all available complexes, and thus possesses rich pre-trained information. Our work involves fine-tuning these types of models on a subsequent antibody dataset. It is precisely because pre-training on the entire PDB dataset is very costly that "repurposing" already pre-trained models is much more convenient and cheap.
>
> > **Q4: Include an AF3-like baseline and necessary of sequence codesign modification**
>
> - Thank you for your question. We would like to clarify that sequence co-design is the main work of this paper, which transforms a structure prediction model into a generative model, rather than just fine-tuning a structure model on antibody data. Therefore, the sequence codesign modification is necessary.
>
> - On the other hand, AF3 itself, as a structure prediction model, does not have the capability to generate sequences in co-design. So we can only compare it on the structure prediction task. In fact, in Table 4, we have already listed the comparison between our fine-tuned model and an AF3-like structure prediction model like Boltz-1 itself, where the model with "Fine-tuned?" as '✗' is Boltz-1. It can be found that our model achieves better results on CDR-masked input and maintains comparable accuracy on CDR-unmasked input.
>
> - Moreover, to demonstrate that our fine-tuned model still maintains good performance on general protein structure prediction, we conducted an additional evaluation. For this test, we selected a set of 128 protein-only complexes from the original Boltz-1 test set. We present the comparison of the relevant performance metrics below. This comparison shows that our fine-tuned model's capability is well-preserved.
>
> | Method   | TM-score | RMSD(A) | DockQ>0.23(%) |
> | -------- | -------- | ------- | ------------- |
> | Boltz-1  | 77.43    | 9.55    | 35.16         |
> | MFDesign | 75.24    | 10.79   | 34.38         |
>
> > **Reference:**
>
> - [1] Zhu, Ren, et al. "Antibody Design Using a Score-based Diffusion Model Guided by Evolutionary, Physical and Geometric Constraints." ICML (2024).
>
> ---
>
> We hope this response could answer your questions and address your concerns, looking forward to receive your further feedback soon.

---

> ### Comment · Reviewer_unJA · 2025-08-01
> **Q2**
>
> I guess my Q2 was a little different, AF3 is a sturcture prediction module trained on the PDB that happens to have a conditional (sequence conditioned) diffusion model trained as part of the process (an addition to AF2, which probably improved a lot of diversity related issues in comparisons), I understand you repurpose the conditional diffusion module within AF3-like models and adapt it to do codesign, and that this model has been trained on the PDB, however my question was more, why not use a pretrained diffusion model thats trained to generate designs (not a structure prediction approach with codesign adaptation) and then finetune this on antibodies?

---

> > ### Author Response · Authors · 2025-08-03
> >
> > Thank you for your prompt response. We hope our additional clarifications below will better address your concerns.
> >
> > > **Pretrain a diffusion-based generative model using all PDB and finetune on antibody dataset**
> >
> > - Thank you for the opportunity to clarify. We agree that finetuning a generation-focused model is a promising direction, though it differs from our work. Our rationale for adapting a folding model like AlphaFold3 rather than a standard generative model is as follows:
> >   - For general protein generative models, most architectures are multi-stage. For example, one might first use RFDiffusion to generate a backbone structure, then use ProteinMPNN to generate the sequence, and finally use RoseTTAFold for folding to validate and screen the results. Another representative work is Chroma. While it integrates structure and sequence generation into a unified architecture, it still follows a sequential process internally—determining the backbone first, then designing the sequence and side-chain. In such paradigms, the generation of sequence and structure is decoupled and handled in a step-wise manner.
> >    In contrast, antibody design primarily utilizes end-to-end co-design for simultaneous structure and sequence generation. As this is our specific task, most general protein generative models, which are not built for true simultaneous co-design, are unsuitable for direct finetuning. Models that can perform this task, rather than in a step-wise manner, are exceptionally rare.
> >   - Furthermore, we believe that in the field of antibody design, a multi-stage paradigm has significant limitations compared to a co-design approach:
> >     - First, the quality of the final sequence is entirely dependent on the quality of the pre-generated backbone. A suboptimal backbone, even if globally plausible, will hinder the design of a high-quality sequence. In other words, errors from the early stages will propagate irreversibly.
> >     - Second, it decouples the reciprocal sequence-structure relationship. For a CDR, a single amino acid change can trigger critical backbone shifts to optimize contacts. A multi-stage model cannot capture this bidirectional feedback because the backbone is frozen before sequence design.
> >     - Finally, regarding computational cost, because multi-stage approaches require sequential design of sequence and structure, plus the use of a folding model for validation, their computational cost and runtime significantly exceed those of co-design models.
> > - Therefore, these reasons fully explain why we did not adopt the approach of finetuning existing generative models. **In fact, one of our baselines, RFAntibody, is a model obtained by finetuning a generative model on antibody data.** It uses an antibody-finetuned RFdiffusion for backbone design, a ProteinMPNN for sequence design and an antibody-finetuned RoseTTAFold2 for final simulation and screening. In our supplementary results (Tables S2, S3, S4, and S5), we indeed have demonstrated that our method achieves higher accuracy and fully comparable binding energy metrics, indicating that our improved accuracy can be translated into high-quality designs.
> > - Next, we would like to explain why, within the co-design setting, we chose to adapt a structure prediction model like AlphaFold3 instead of a generative model like RFDiffusion. There are several key reasons:
> >   - First, AlphaFold3 is trained to understand sequence-structure synergy, as it predicts structure from sequence. In contrast, RFDiffusion's task of generating structures from noise means it inherently lacks this rich information, making a prediction model like AlphaFold3 more advantageous for our co-design task.
> >   - Second, AlphaFold3's architecture provides intermediate representations like $a _{i}$, ideal for simultaneous sequence and structure generation. RFDiffusion's architecture, focused on generating coordinates from noise, is not designed to produce such sequence-aware representations. Adapting it would require substantial re-engineering with uncertain outcomes.
> >   - Finally, AlphaFold3 can achieve full-atom structure prediction, whereas standard RFDiffusion can only design backbone atoms (since it lacks sequence design functionality, it cannot generate side chains). AlphaFold3's full-atom prediction capability is more beneficial for our sequence design and generation.
> > - In summary, we chose an architecture suited for an end-to-end co-design strategy, which we believe is more direct, effective, and lower-cost than common multi-stage methods. We still agree that finetuning multi-stage general protein generative models is a very promising direction, as it might enable more diverse and customized designs. Furthermore, that approach can more easily support full-atom structure design (because the step-wise process allows for side-chain generation after the sequence is determined, which is non-trivial in a co-design setting). We consider this a valuable avenue for future research.

---

> > > ### Comment · Reviewer_unJA · 2025-08-05
> > > **Thank you**
> > >
> > > As before, I would like to thank the authors for taking the time to explain/clarify my concerns. I maintain my view that the work is both interesting and impactful, and my questions have been clarified. I will keep my positive score (it was a 5) as I believe this work should be accepted.

---

> > > > ### Author Response · Authors · 2025-08-06
> > > >
> > > > We sincerely thank you for your time and for engaging with our rebuttal. We are very pleased to know that our explanations have clarified your questions. We truly appreciate your positive assessment of our work as "interesting and impactful," and we are deeply grateful for your continued support. Your encouragement means a great deal to us.

---

### Comment · Area_Chair_7tsg · 2025-08-05

Dear reviewers,

Thanks for your contribution to the reviews, and also thanks for those who have make responses yet.
The authors have put rebuttal to the reviews, please look at the rebuttal and make responses accordingly. Note that engaging the discussion is important at this phase.
Please make sure that you give necessary comments to the rebuttal and then make acknowlegement, therefore the authors can have a better understanding whether their rebuttal help to your concerns.

Best,
AC

---

> ### Comment · Area_Chair_7tsg · 2025-08-06
>
> Dear reviewers,
>
> Please don't forget to give necessary feedback to the rebuttal, and also don't foreget to make acknowledgement if you haven't done.
>
> AC.

---

### Note · Authors · 2025-08-12

We sincerely thank the chairs and reviewers for their time, feedback, and constructive suggestions.

In the **initial review (scores: 5, 4, 4, 3)**, we were delighted that reviewers acknowledged our work's novelty (unJA, jMg6), significance (ppEN, jMg6), and its potential for making "breakthrough strides" (unJA) in antibody design. Key technical contributions, such as the repurposing of a large foundation model and our rigorous, well-filtered evaluation protocol (jMg6, RhHw), were highlighted as major strengths.

------

During the discussion period, we focused on addressing the primary concerns:

- **Clarification of Core Contribution & Novelty:** We have refined the manuscript to more clearly articulate our core contribution: the transformation of a predictive model (AF3/Boltz-1) into a generative co-design framework, clarifying why this is a necessary and novel step beyond the baseline model's capabilities.

- **Strengthened Evaluation & Rigor:** We have significantly strengthened our evaluation by adding statistical uncertainty estimates (SEMs) to demonstrate significance, incorporating a second benchmark (RAbD) with retrained baselines, and providing a detailed computational cost analysis for practical context.
- **Expanded Discussion on Limitations & Scope:** In response to feedback, we expanded our discussion on the model's limitations for a more balanced perspective. This includes addressing the backbone-only generation approach, the practical context of in-silico metrics, and clarifying the hardware-related nature of the >2k residue limit for full transparency.

------

**We are thankful for the constructive dialogue during the discussion period. We were heartened to see our additional work and clarifications resonate with the reviewers, prompting Reviewers jMg6 and RhHw to increase their scores in favor of acceptance.** Both reviewers noted that the added analyses substantially strengthen the manuscript, and we commit to incorporating all revisions into the final version.

We hope this work can introduce new vitality to the antibody engineering and machine learning communities. We believe that by leveraging powerful foundation models, we can continue to unlock new frontiers in therapeutic design.

Once more, we extend our sincere gratitude to the chairs and reviewers. Their guidance has been instrumental in elevating the quality of our work.

---

### Decision · Program_Chairs · 2025-09-17

**Decision:**

Accept (poster)

**Comment:**

This paper proposes to repurpose AF3-like protein folding models for antibody sequence-structure co-design. Specifically, the framework uses a sequence diffusion head and finetune the protein folding model. With experiments on antibody sequence-structure co-design tasks, the results show great improvements.

Reviewers all agree the paper is well presented and well motivated, more importantly, the performance show strong strengths compared to existing works, especially that the results can compare to Chi or related models. As for improvements, the authors are encouraged to add more discussions and revised points as discussed in rebuttal.